# Addressing human-tiger conflict using socio-ecological information on tolerance and risk

Matthew J. Struebig[1], Matthew Linkie[1,2], Nicolas J. Deere[1], Deborah J. Martyr[3], Betty Millyanawati[3], Sally C. Faulkner[4], Steven C. Le Comber[4], Fachruddin M. Mangunjaya[5], Nigel Leader-Williams[6], Jeanne E. McKay[1] & Freya A.V. St. John[1,7]

Tigers are critically endangered due to deforestation and persecution. Yet in places, Sumatran tigers (*Panthera tigris sumatrae*) continue to coexist with people, offering insights for managing wildlife elsewhere. Here, we couple spatial models of encounter risk with information on tolerance from 2386 Sumatrans to reveal drivers of human–tiger conflict. Risk of encountering tigers was greater around populated villages that neighboured forest or rivers connecting tiger habitat; geographic profiles refined these predictions to three core areas. People's tolerance for tigers was related to underlying attitudes, emotions, norms and spiritual beliefs. Combining this information into socio-ecological models yielded predictions of tolerance that were 32 times better than models based on social predictors alone. Pre-emptive intervention based on these socio-ecological predictions could have averted up to 51% of attacks on livestock and people, saving 15 tigers. Our work provides further evidence of the benefits of interdisciplinary research on conservation conflicts.

[1] Durrell Institute of Conservation and Ecology (DICE), School of Anthropology and Conservation, University of Kent, Canterbury CT2 7NR, UK. [2] Wildlife Conservation Society – Indonesia Program, Jl. Tampomas No. 35, Bogor 16151, Indonesia. [3] Fauna and Flora International – Indonesia Programme, Kampus Universitas Nasional (UNAS), Jl. Sawo Manila No. 61, Pejaten, Jakarta 12550, Indonesia. [4] School of Biological and Chemical Sciences, Queen Mary University of London, London E1 4NS, UK. [5] Faculty of Biology, Universitas Nasional (UNAS), Jl. Sawo Manila Pejaten, Pasar Minggu, Jakarta 12520, Indonesia. [6] Department of Geography, University of Cambridge, Downing Place, Cambridge CB2 3EN, UK. [7] School of Environment, Natural Resources and Geography, Bangor University, Bangor LL57 2UW, UK. Correspondence and requests for materials should be addressed to M.J.S. (email: m.j.struebig@kent.ac.uk)

Conservation science is often hindered by disciplinary boundaries[1]. Consequently, despite benefits for management, research exploring links between ecological and social systems is limited[1–3]. This is particularly important when addressing human–wildlife conflicts[4,5], as truly interdisciplinary socio-ecological research is challenging[2], resulting in ecological and social components being frequently studied independently[6–10]. Situations involving mammalian carnivores exemplify this problem, as many are highly threatened, heavily persecuted and pose a public threat[11]. Within social–ecological frameworks invoked to explain human–carnivore interactions factors associated with people's risk of attack, and/or their motivation to retaliate are central[4,12,13]. Yet, much research has been driven by natural scientists seeking to predict and map risk from wildlife encounters to target people or problem animals[7,14]. As statistical advances and the inclusion of social data improve the practical value of these methods, it is pertinent to move away from single-model techniques[6,7,15], and ensure that socio-ecological insights are translated into mitigation efforts[5].

Understanding people's degree of tolerance for wildlife is key to managing dangerous and/or damage-causing species, and promoting coexistence[12]. Tolerance is a passive concept requiring no action, whereas intolerance may be expressed through actions such as killing of, or opposition towards, certain species[16,17]. Tolerance may take attitudinal and behavioural forms[13]: people may have negative attitudes towards animals, which are then reflected in the behavioural act of killing[18]. These negative perceptions need not result in such extreme acts, nor are they always based on history of experiencing harm[19]. However, they can influence the degree to which people enable persecution or support conservation[20]. Therefore, applying psychological theory to ecological questions of human–wildlife conflict helps us understand how judgements about wildlife are formulated and how they relate to people's tolerance. Such insights are valuable for designing interventions[2].

People process information using analytical and experiential systems[13,20]. The analytical is deliberative and involves cognitively burdensome rational decision making (e.g. one deliberates knowledge concerning animal numbers before deciding how to act). The experiential is intuitive and automatic, and is largely driven by affect; the instant reaction one has to a stimulus (e.g. how scared one is of an animal[20]). Like affect, trust and norms operate as heuristics, or decision-making shortcuts[13,21], and can reduce people's perception of risk. Greater social trust—the willingness to rely on those responsible for making decisions[22] (e.g. protected area management authorities)—should lead to greater acceptance of wildlife[13]. Deviation from social norms— the shared expectations about appropriate actions that influence people's behaviour in groups—is known to deter rule-breaking in conservation due to feelings of shame or exclusion[10,21,23]. Together, these social–psychological factors are a pervasive influence on people's decision making, and should therefore play a large role in shaping tolerance of wildlife.

The need for socio-ecological information in conservation conflict mitigation is particularly urgent for tigers, which are flagship species and cultural icons for biodiversity and rainforest protection[11,24]. Tigers are on the brink of extinction, having undergone steep declines as a result of habitat loss, retaliatory killings and poaching[25]. Despite this trend, people continue to coexist with tigers in Sumatra, with the tiger population comprising ca. 500 individuals, around 20% of global numbers in the wild[25]. Tigers have struggled to survive in areas converted to large-scale agriculture, but continue to flourish in forests bordering smallholdings, despite ongoing encounters with people[26].

Tolerance of Sumatran tigers has previously been explained by the acceptance of Islam, which prohibits eating animals that hunt with claws, as well as other belief systems held by indigenous farming communities[27–31]. For example, the Kerincinese and Minangkabau people have lost kin to tigers through the centuries, but have long-standing spiritual connections with animals, including that ancestral souls are embodied within tigers, which serve as guardians of customary laws[32,33]. Some also believe individuals can assume the form of a 'weretiger', which can cause havoc unless habituated[27,28,31]. These spiritual belief systems, coupled with ongoing monitoring of tigers and their encounters with people, present a unique opportunity to investigate how these factors might foster tolerance and coexistence with dangerous wildlife.

Here we develop a socio-ecological approach to the study of human–wildlife conflict be integrating spatial models of encounter risk with wildlife perception questionnaire data from Kerinci Seblat, a stronghold for Sumatran tigers[26,34]. The Kerinci Seblat landscape comprises 13,800 km$^2$ of mountainous national park and surrounding forest and farmland. Since 2000 up to six Tiger Protection and Conservation Units have worked around the park, responding to incidents involving tigers and people, and conducting de-snaring patrols inside the forest[35,36]. Units cover >2300 villages across ca. 10,000 km$^2$ of remote countryside, and comprise ranger patrols, conflict-resolution personnel responding to public demands and an informant network that have contributed to poacher arrests. We mapped encounters from Unit reports and local media between 2000 and 2013, and grouped this information into four encounter types according to the Indonesian Government's Problem Tiger Management Strategy[37]: (i) 106 sightings, (ii) 83 attacks on livestock, (iii) 12 attacks on people and (iv) 27 removals of tigers, typically by snare or poison, with individual incidents occurring at least 1 month apart in the same village. Most incidents were resolved by mitigation techniques such as noise deterrents, but 35% of cases escalated to another encounter, including at least four tigers being intentionally killed following sightings (Fig. 1a). Although Units remained active across the landscape, reports from villagers ceased in early 2014 as poachers infiltrated the area in response to increased demand for tiger skins[35]. In the continued absence of encounter data, it is important to prioritise interventions to reduce casualties of livestock, people and tigers, and learn lessons that could be applied to other socio-ecological contexts.

We first reveal how landscape variables influence human–tiger encounters and can be used to predict the spatial patterns of encounter risk. We make use of traditional landscape ecological models, as well as geographic profiling, a technique developed in criminology to help prioritise search lists of serial crime suspects, with recent applications in epidemiology and ecology[38,39]. We then utilise a questionnaire survey of 2386 villagers across the landscape[16], and demonstrate how factors underpinning human decision-making explain variation in tolerance towards tigers. Finally, given limited resources in conservation, we integrate the ecological and social information on risk and tolerance to help prioritise interventions, with a view to avoiding losses to people, or tigers from the wild.

## Results

**Spatial risk of tiger encounter.** An ensemble model of encounter risk combined information from three predictive algorithms with good discriminatory power. The weighted average area under the curve (AUC) across the three models was 0.78, comprising a random forests (RFs) model (AUC = 0.85), generalised additive model (GAM) (0.76) and generalised linear model (GLM) (0.72); discriminatory power of a generalised boosted and support vector machine models (SVMs) being relatively poor (AUC <0.70). According to this consensus approach, the probability of

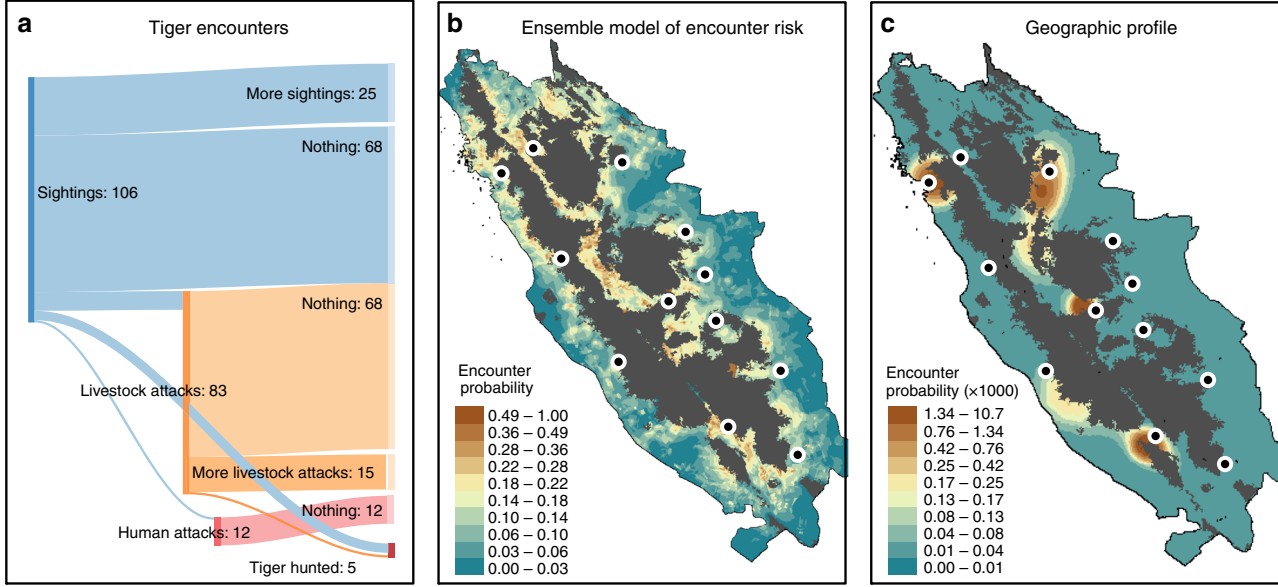

**Fig. 1** Tiger encounters over Kerinci Seblat between 2000 and 2013. **a** Sankey diagram of the progression of human–tiger encounters from sightings (blue), through to attacks on livestock (orange) and people (red), and tigers hunted. Encounters are linked if they occurred <6 months apart in the same village. The overall number of tigers hunted was 27, of which 5 were linked to previous encounters. **b** Risk of encounter predicted by an ensemble of binomial models, or **c** a geographic profile based on all 228 reported encounters 2000–2013. Locations of the study areas are shown by points. Probabilities across the entire geographic profile sum to 1

encountering a tiger (via sightings, attacks on livestock or people, or retaliatory killings, Fig. 1a) was most associated with distance to rivers, distance to forest, connectivity of the landscape for tiger movement, human population density and tiger occupancy ($R_{Pearson}$ values <0.1 in correlations between final model and null model without variable investigated) (Supplementary Table 1). Other covariates included in the models were weak predictors, including population density of farmers, percent forest in the landscape and distance to roads. Risk was relatively high across much of the region, especially near forest, but the top percentile of risk probabilities were restricted to scattered localities across the landscape (Fig. 1b). An alternative ensemble, based on models utilising sightings alone, generated a near identical outcome (weighted average AUC across the three models = 0.75: RFs AUC = 0.82, GAM AUC = 0.72, GLM AUC = 0.72. Supplementary Fig. 3).

**Geographic profiling of tiger encounters**. In contrast to the ensemble risk model, a geographic profile based on all encounter data revealed four main clusters of activity, with the top 10% of probabilities limited to three areas (in Merangin, Pesisir Selatan and Lebong; Fig. 1c). A profile based on tiger sightings alone produced a similar spatial prediction, and was able to identify 40% of the subsequent attacks and incidents where tigers had been killed by searching <15% of the region, and 80% of these encounters by searching just 30% (Gini coefficient = 0.644, Supplementary Fig. 4), thus validating model performance. In other words, a large number of tiger attacks tended to be restricted to a few geographic areas, and this same pattern was evident from reports of sightings. Therefore, sighting data from a few specific areas are more informative to help alert response units of potential future incidents before they escalate to injury or loss of life. Moreover, interventions could also be pre-emptive, being based on reports of tiger sighting, and focused in a limited number of areas before an attack takes place.

**Defining tolerance towards tigers**. We surveyed 2386 people across Kerinci Seblat. Most were male (73.9%), and the mean age was 43.8 years (SE = 0.26). Respondents self-identified as Minangkabau (45.4%), Melayu (32.5%), Javanese (7.1%), Rejang (6.5%), Kerincinese (2.9%) or other ethnic group (5.5%)[16]. Less than 1% had been personally injured by a tiger or lost livestock in the previous year, but many could recall stories of tiger encounters in the landscape, including attacks elsewhere. We measured tolerance as the capacity for people to accept wildlife (*sensu*[16,40]) by asking respondents whether they would prefer the tiger population to be reduced/eradicated, stay the same or increased. Overall, 28.1% of respondents opted for reducing/eradicating tigers, 48.0% for the current population level and 19.4% for an increase. People who did not know what type of change they wanted to see in the size of the tiger population (4.5%, *n* = 107) or who had missing data (1.2%, *n* = 30) were excluded from subsequent modelling.

The questionnaire examined the role of social–psychological characteristics of human behaviour (Fig. 2), as well as age, sex and ethnicity that might influence tolerance towards tigers (Supplementary Table 3). A multinomial logistic regression model was used to examine the relationships between these variables and tolerance amongst respondents (Tables 1 and 2). This model was then repeated with risk covariates included to determine the extent to which tolerance was defined by both ecological and social factors (i.e. with the expectation that combining information in this way would improve model performance). The model was only slightly improved by adding the ecological covariates from the encounter risk models to the social covariates (reduction in Akaike's information criterion with a correction (AICc) = 0.14), or by including output probabilities from the ensemble model of encounter risk (reduction of 0.58), but was improved substantially when combined with the geographic profile measure of risk (reduction of 6.96; Table 1). Examination of evidence ratios based on ΔAIC values[41] indicates that the model combining social covariates

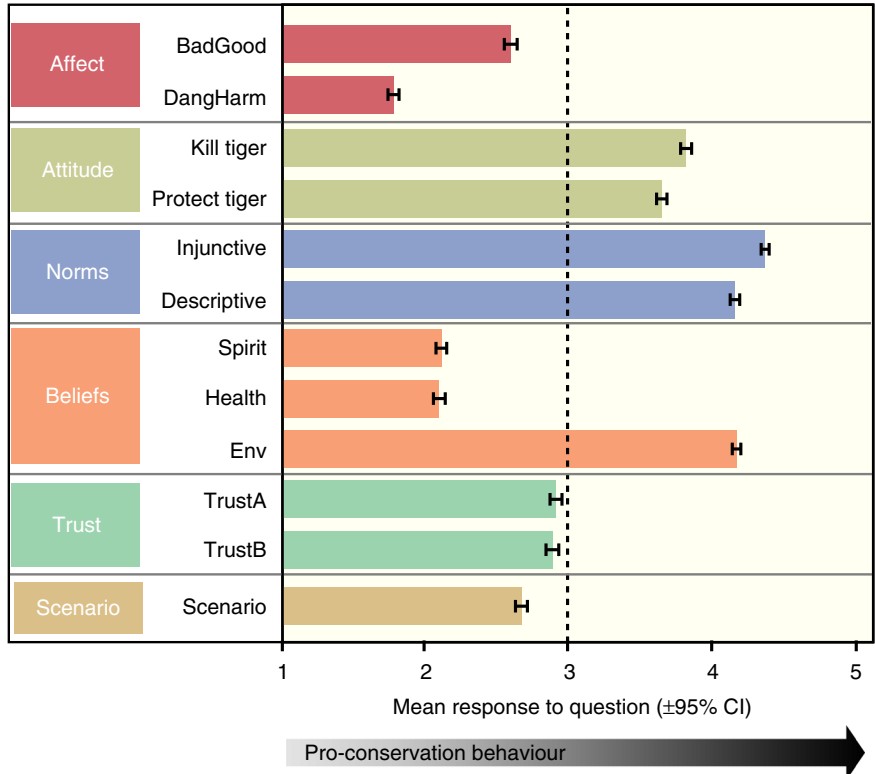

**Fig. 2** Distribution of social variables reported by respondents in the tolerance questionnaire. Mean responses to questions on affect, attitude, norms, beliefs, trust and management scenarios concerned with tigers across 2386 villagers in Kerinci Seblat, Sumatra. Responses are rescaled from the original questions so that scores higher than three (neutral) indicate pro-conservation values. Error bars represent 95% confidence intervals from the 2386 responses

| Table 1 Performance of tiger tolerance models with and without landscape covariates or measures of risk | | | | |
|---|---|---|---|---|
| **Model and covariates** | **AICc** | **ΔAICc** | **Log-like** | **K** |
| *Social predictors + geographic profile*: Kill_tiger + Protect_tiger + Injunctive + Descriptive + BadGood + DangHarm + Spirit + Health + Env + TrustB + Scenario + Age + Sex + Ethnicity + GP | 3760.59 | 0 | −1846.30 | 34 |
| *Social predictors + ensemble risk probability*: Kill_tiger + Protect_tiger + Injunctive + Descriptive + BadGood + DangHarm + Spirit + Health + Env + TrustB + Scenario + Age + Sex + Ethnicity + Prob_conf | 3766.97 | 6.38 | −1848.79 | 34 |
| *Social plus landscape covariates from ensemble risk model*: Kill_tiger + Protect_tiger + Injunctive + Descriptive + BadGood + DangHarm + Spirit + Health + Env + TrustB + Scenario + Age + Sex + Ethnicity + dis_for + dis_rds + dis_riv + connect + farmers + pop_grd + occupancy | 3767.41 | 6.82 | −1839.70 | 44 |
| *Social predictors only*: Kill_tiger + Protect_tiger + Injunctive + Descriptive + BadGood + DangHarm + Spirit + Health + Env + TrustB + Scenario + Age + Sex + Ethnicity | 3767.55 | 6.96 | −1851.77 | 32 |

Models are presented in order of performance according to Akaike's information criterion corrected for small sample sizes (AICc). The ΔAICc indicates the difference in AIC relative to the top performing model. Two measures of encounter risk were explored: an ensemble model combining the outputs of three presence–absence algorithms (Prob_conf), and a geographic profile (GP). A third model incorporated the landscape predictors utilised in the ensemble predictor of risk. Social covariates were identical throughout. All social covariates and risk scores (probability of conflict, or geographic profile) were entered as fixed effects. Data sources, covariate abbreviations and analyses are described in the Methods and Supplementary Table 3

and geographic profile information performed around 24 times better than the model using the ensemble model risk probabilities, and 32 times better than the one limited to social predictors alone. People's tolerance towards tigers was therefore driven by how likely they were to have encountered a tiger in the past (i.e. risk perception informed by past events), as well as their beliefs and perceptions.

**Relative importance of beliefs in defining tolerance**. In addition to age, ethnicity was not selected in our models as an important predictor of people's tolerance towards tigers (Table 2). Rather,

underlying psychological factors, including attitudes, human emotion, and beliefs associated with overall spiritual well-being, were the strongest significant predictors of people's connections with tigers overall, as evidenced by large model-averaged $\beta$ coefficients and variable importance values (Table 2). Tolerant responses among people were driven by positive attitudes towards protecting, affective responses—whether the instant reaction one has to a tiger is positive—and perceived importance of tigers for spiritual well-being. Similarly, if respondents considered it unusual for people to catch tigers in their village (i.e. descriptive norms), they were more likely to be tolerant, echoing trends in other carnivore conflict systems[10,42]. Nevertheless, our measure

**Table 2 Multinomial logistic regression model describing predictors of people's tolerance to tigers**

| Predictors of tolerance to tigers | β | SE | z value | P value | Importance |
|---|---|---|---|---|---|
| *Increase vs. eradicate/reduce* | | | | | |
| **Intercept** | **−14.07** | **0.87** | **16.16** | **<0.001** | **-** |
| **Protect tiger** | **0.770** | **0.10** | **7.55** | **<0.001** | **1.00** |
| **Kill tiger** | **0.619** | **0.10** | **6.11** | **<0.001** | **1.00** |
| **BadGood** | **0.580** | **0.08** | **7.29** | **<0.001** | **1.00** |
| **Spirit** | **0.542** | **0.09** | **5.76** | **<0.001** | **1.00** |
| **Descriptive** | **0.402** | **0.10** | **3.66** | **<0.001** | **1.00** |
| **Scenario** | **0.401** | **0.10** | **3.97** | **<0.001** | **1.00** |
| **Env** | **0.381** | **0.01** | **2.53** | **0.011** | **1.00** |
| **Health** | **0.381** | **0.08** | **4.77** | **<0.001** | **1.00** |
| **Geographic profile** | **−0.245** | **0.09** | **2.85** | **0.004** | **1.00** |
| **TrustB** | **0.159** | **0.07** | **2.19** | **0.028** | **1.00** |
| DangHarm | 0.114 | 0.09 | 1.32 | 0.19 | 1.00 |
| **Sex: Male** | **0.929** | **0.20** | **4.75** | **<0.001** | **1.00** |
| Age | −0.006 | 0.01 | 0.88 | 0.37 | 0.56 |
| *Keep same vs. eradicate/reduce* | | | | | |
| **Intercept** | **−6.55** | **0.60** | **10.95** | **<0.001** | **-** |
| **Protect tiger** | **0.577** | **0.07** | **8.81** | **<0.001** | **-** |
| **Kill tiger** | **0.328** | **0.06** | **5.15** | **<0.001** | **-** |
| **BadGood** | **0.582** | **0.06** | **9.10** | **<0.001** | **-** |
| Spirit | 0.134 | 0.08 | 1.61 | 0.108 | - |
| **Descriptive** | **0.403** | **0.08** | **5.04** | **<0.001** | **-** |
| Scenario | 0.084 | 0.07 | 1.17 | 0.241 | - |
| Env | 0.138 | 0.08 | 1.70 | 0.089 | - |
| Health | 0.120 | 0.07 | 1.75 | 0.080 | - |
| **Geographic profile** | **−0.132** | **0.05** | **2.44** | **0.015** | **-** |
| **TrustB** | **−0.168** | **0.06** | **2.92** | **0.003** | **-** |
| DangHarm | −0.103 | 0.07 | 1.38 | 0.168 | - |
| **Sex: Male** | **0.356** | **0.13** | **2.79** | **0.005** | **-** |
| Age | −0.001 | 0.003 | 0.28 | 0.78 | - |

Tolerance is defined as a respondent's preference for the tiger population level with the reference category as 'Eradicate/Reduce'. Model-averaged coefficients (β) and standard error (SE) indicate the strength of selection for or against a covariate with positive values indicating selection for and negative against. For psychological variables, a positive coefficient implies a more pro-conservation value; for sex it indicates that men are more likely than women to support increase to tiger population or keep it the same compared to eradicate; for the geographic profile, the negative coefficient implies the further respondents are from cluster of risk the more inclined they are to support increase in tiger population or keep the same, rather than eradicate. Significant predictors are highlighted in bold. All social covariates and risk scores (geographic profile) were entered as fixed effects. For covariate abbreviations see Methods and Supplementary Table 3

of injunctive norms was not retained in the averaged model, implying that, in their behaviours towards tigers, people are unlikely driven by this form of social pressure.

People's trust in management authorities (national park staff and Indonesia's Nature Conservation Agency) was inconsistently related to tolerance (Table 2): those opting for an increase, rather than reduction, in tiger numbers had higher trust in the authorities. However, trust was weakly and negatively related to tolerance for the current, compared to reduced, tiger population size, suggesting that whilst people trust the authorities to keep them safe from dangerous animals, trust alone does not generate tolerance.

**Prioritising intervention based on risk and tolerance data**. Using a simple framework coupling the ensemble measure of risk with the measure for tolerance (Fig. 3a; Supplementary Table 4), 44% of our surveyed villages would be identified as high priority for intervention (i.e. above-average risk and below-average tolerance), and 11% as low (below-average risk). At least 40 attacks

of livestock and people were reported from the high priority settlements in the 13 years of records; 54% of all attacks in our study villages. Prior to 2013, the removal of at least 15 tigers (65% of all reported) may have been avoided if these villages had been prioritised earlier, a highly significant number considering the rarity of tigers and the size of the Kerinci Seblat landscape.

If prioritisations were made based on the geographic profile measure of risk coupled with tolerance data, response units would first be directed to 8% of villages in high risk, low tolerance areas (Fig. 3b; Supplementary Table 4). Only 15% of attacks and 35% of tigers hunted would have been highlighted via this approach, but efforts would have only been focussed in six villages, allowing intervention efforts to be redeployed elsewhere. Moreover, using the geographic profile would be particularly useful in situations where encounter data are limited to a small number of sightings.

## Discussion

Evidence suggests that applying socio-ecological models to conservation conflicts can be informative and beneficial; such models are rare, but have been increasingly applied in recent years[5]. By providing an application of risk modelling that incorporates geographic profiling as well as ecological and social data, we demonstrate an important interplay between ecological predictors of risk and the social context that drives people to intolerant attitudes and behaviours towards dangerous wildlife. Social models of tolerance that included ecological information on risk were up to 32 times better than using social predictors alone. Furthermore, combining information on tolerance to prioritisations based on risk allows villages to be ranked according to their likelihood of retaliation, allowing valuable mitigation resources to be invested where they are needed most.

Our results suggest that the prevailing anthropological view that attitudes towards wildlife in Southeast Asia are driven by spiritual beliefs unique to specific indigenous ethnic groups[27–29,32] could be oversimplified. Rather, our findings indicate that tolerance towards tigers, in Kerinci Seblat at least, is driven by a number of complex factors that include spirituality, as well as risk of actual attack. Notably, respondents in our landscape were more likely to support an increase in tiger populations if they held beliefs concerning the importance of the animal for spiritual well-being, but this inclination appeared broadly prevalent amongst respondents and independent of their ethnicity (Table 2). We recognise that the long-standing spiritual connections with wildlife inherent to Minangkabau and Kerincinese people may still be important in some areas[31], and that our survey was not implicitly designed to test for differences amongst ethnic groups. Yet from our data at least, attitudes, affect, beliefs and norms across rural Sumatran society, in addition to risk of actual attack, appear more important factors in defining human–tiger interactions over large spatial scales. Whether this has always been the case, or is something that has developed as spiritual relationships have come under pressure from a changing Indonesian society[43] is unclear. Measures of tolerance may better reflect people's actual ability to coexist with a species if linked explicitly to an effect, for example, the probability of increased human–tiger interaction given a specified population increase. However, data to specify such statements are lacking in most human–wildlife systems.

Resolving human–wildlife conflicts is particularly challenging for conservation agencies, because it involves meeting multiple goals for managing threatened species and the people with which they interact. In tiger-range countries, these are often some of the poorest and marginalised people in society. Millions of US dollars in donor funds are spent on in situ tiger conservation annually[24], making it all the more important to direct resources to where they are needed most. Applying our social–ecological prioritisation

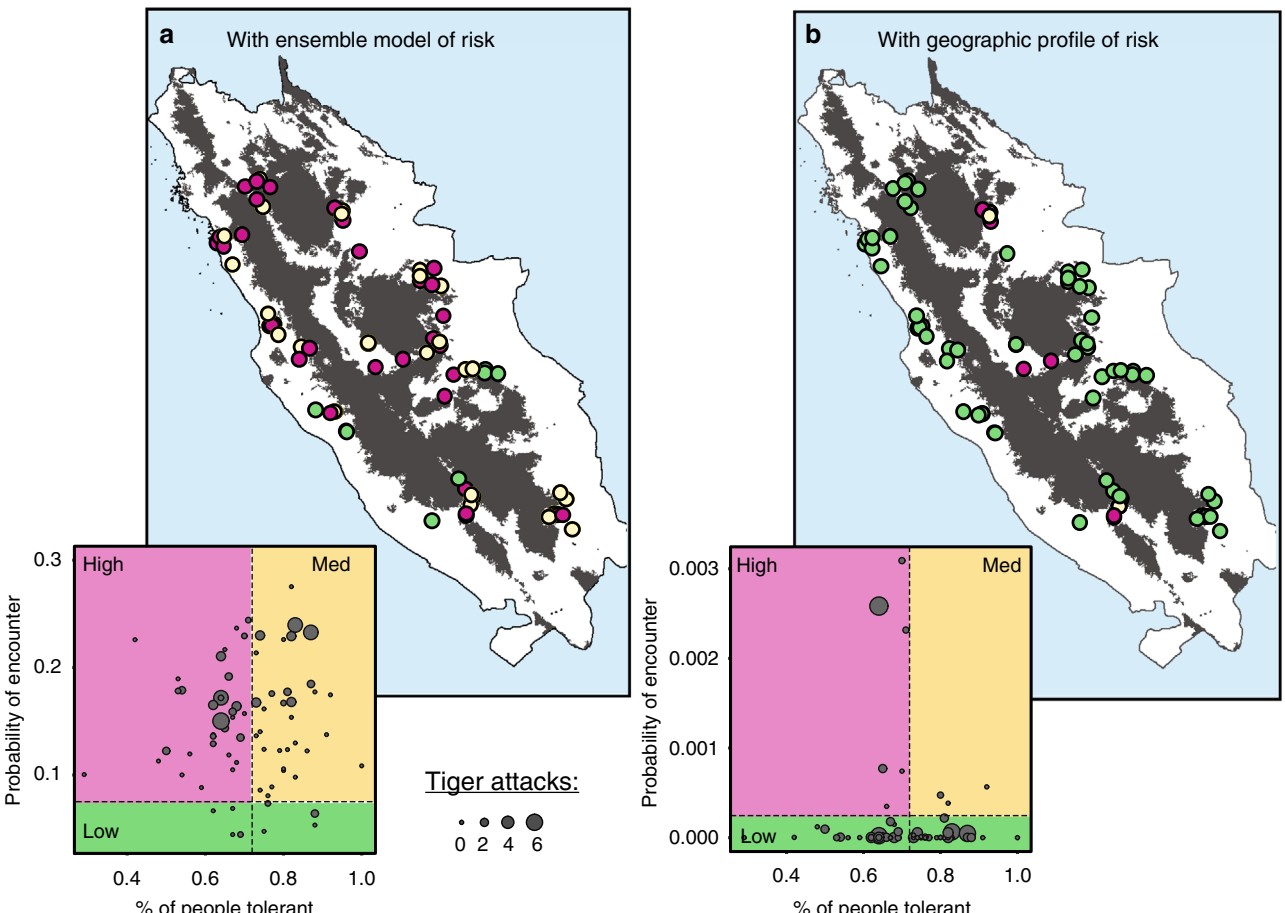

**Fig. 3** Prioritisation of villages for tiger conflict mitigation efforts using tolerance and risk data. The 75 survey villages are partitioned into high (magenta = above median risk; below median tolerance), medium (beige = above median risk; above median tolerance) and low (green = below median risk) priorities using **a** an ensemble of binomial risk models, or **b** a geographic profile. The estimates of encounter risk were based on models utilising all 228 human–tiger encounters; alternative estimates from models using sightings data only provided the same result. Point size is weighted by the number of attacks (on livestock or people) reported in each village

framework assumes that characteristics of individuals surveyed within our study villages are representative of the wider Kerinci Seblat region. It also assumes that our measures of risk adequately describe the region and that the trends from the past 13 years continue, an important assumption given the drop in reported encounters yet spike in poaching since 2013. As there was no discernible change in policy, corruption, public concern, or conservation funding for tigers during the study period[35], poaching trends are unlikely linked to changes in local communities. Indeed poaching rates are known to fluctuate, and are positively correlated with tiger skin prices and the wider condition of the Indonesian economy[35].

The finding that substantial numbers of incidents took place in villages deemed low priority under our framework indicates that factors other than risk or tolerance are at play in Kerinci Seblat, and/or additional variables might improve our predictions. For example, tiger occupancy data were informative to our ensemble encounter models, but were based on past survey data, after which densities are thought to have declined[26]. Incorporating dynamic variables into the ecological models as survey coverage is expanded could potentially improve our predictions. Similarly, there is little information on the availability of prey or accessible livestock, which might also influence tiger encounters[7,8]. While improved ecological information could help strengthen predictive performance of encounter models, many socio-ecological systems are lacking such information, and so a move to geographic

profiling methods based on few locality data could be particularly useful to delineate risk. Despite these shortcomings, our study suggests that attacks by and towards tigers could be averted or reduced by prioritising responses or intervening proactively (e.g. via predator-proof livestock enclosures, compensation payments and de-snaring patrols). We encourage conservation practitioners to expand upon and apply our framework to other socio-ecological systems globally to more fully explore its utility for facilitating human–wildlife coexistence. Although the spiritual connections with wildlife reported from Sumatra are somewhat unique, cultural tolerance is known to enable greater persistence of carnivore species in regions as diverse as India[44] and Ethiopia[45]. Where such beliefs are not widely held, or are outweighed by fear, retaliatory killings increase, as found for jaguars in Brazil[42]. Therefore, the socio-ecological interplay between risk and tolerance permeates many human–wildlife conflict situations, and so locally adapting our framework to these contexts could help avoid further losses to people and some of the world's most endangered species.

## Methods
**Ecological determinants of encounters**. We defined our study landscape by a 5 km buffer around the Kerinci Seblat National Park and adjacent forest reserves, within which the vast majority of encounter records were reported. Within this landscape we compiled the following spatial variables to help predict areas of high risk of tiger encounter: distance to rivers (dis_riv), distance to roads (dis_rds), distance to forest (dis_for), percent forest cover (for_cov), human population

density (pop_grd), farmer population density (farmers), tiger occupancy (occupancy), and tiger connectivity (connect) (Supplementary Tables 1 and 2 and Supplementary Figs. 1 and 2). Connectivity was derived by electronic circuit theory to simulate tiger movements across areas of variable resistance[46], based on land cover, topography and distance to rivers (Supplementary Note 1). Tiger occupancy, based on surveys implemented between 2007 and 2009[34], was then spatially interpolated to non-surveyed areas. To account for possible georeferencing errors, we extracted mean values of each spatial variable from a 3.25 km radius (the average village size) of each point. Variables were scaled and centred before analysis, and collinearity was assessed using Pearson's correlation coefficients and variance inflation factors (VIFs), with all predictors used having $|r| < 0.7$ and VIF $<3$. Interaction terms were not included due to the limited number of samples.

**Spatial models of encounter risk.** To determine the probability of human–tiger encounters, we produced an ensemble model that combined the predictions of up to five different spatial algorithms that predicted presence–absence: GLMs, GAMs, RFs, SVMs and boosted regression trees. All 228 incidents were assigned a value of '1', and 10,000 localities were randomly drawn from the background and assigned '0'. We drew our pseudo-absence data from the full farmland extent since the encounter database indicated that incidents were possible across the whole landscape over the 13 years of records. To evaluate the predictive performance of each algorithm, we used a random subset of 70% of the data to calibrate the model, and the remaining 30% for evaluation, using the area under the relative operating characteristic curve (AUC). This was replicated ten times to calculate a mean AUC of the cross-validation. Predictions from models with moderate–good fit (AUC $>0.70$) were included in the final ensemble, and the weighting of each algorithm prediction was based on its true skill statistic[47]. The average contribution of each environmental variable across all selected models was calculated as a Pearson's coefficient of the correlation between fitted values and predictions where each variable was permuted via a randomisation procedure, that is, low coefficients correspond to high variable importance[48]. Ensemble modelling was implemented in the R package SSDM[49].

**Geographic profiling encounters.** Unlike the five algorithms used in our ensemble modelling, geographic profiling is based solely on spatial information, and does not require parameters from other spatial layers or the study system to make predictions. The technique is used to estimate probable sources of spatial data (in this case human-tiger encounters), and is particularly robust when very few data points are used, or when there are an unknown number of sources (e.g. multiple tiger territories, or hunter groups), far outcompeting other spatial statistics across a wide range of contexts[50]. Geographic profile models share a distance decay feature, which specifies limitations to travel away from key anchor points (e.g. a tiger territories; livestock areas).

We produced a geographic profile of tiger-encounter data using the Dirichlet Process Mixture (DPM)[38,51], implemented in the R package Rgeoprofile version 2.1.0 (https://github.com/bobverity/Rgeoprofile). Conceptually the DPM model comprises two parts: first, locations (i.e. tiger encounters) are partitioned into $n$ clusters (the number of which does not need to be specified in advance) with neighbouring encounters most likely included within the same cluster. Then, source locations are estimated from the clustering using a Gibbs sampler to alternate between these steps thousands of times within a Markov Chain Monte Carlo framework until the algorithm converges on the posterior distribution of interest. The DPM model fits $\sigma$, the standard deviation of the bivariate normal distribution around the source(s), in km units, from the data. Duplicate points in the dataset resulted in a tendency to fit very small values of $\sigma$. Therefore, we first ran a model from the 105 unique points, and used the resulting $\sigma$ of 27 km to run the full dataset, using 50,000 samples and 10 chains with a burn-in of 10,000.

To evaluate model performance, we ran a third model based solely on tiger sightings to predict where future tiger encounters (i.e. attacks and poaching) might occur. Model output was assessed using hitscores—the proportion of the study area searched by the model before a source (i.e. encounter) is located. Overall model performance was assessed using a Gini coefficient. In this case, we compared the proportion of tiger encounters identified using the sightings data alone, to the proportion of the total area searched. A Gini coefficient of one would have a perfect search strategy—the higher the Gini coefficient the more effective the search strategy, and more accurate the geographic profile.

**Village surveys to quantify tolerance.** A questionnaire-based survey was implemented alongside a study investigating socio-cultural attitudes towards wildlife in Sumatra[16]. To ensure we represented the full range of views regarding tigers, we stratified questionnaire sampling across 11 equal (ca. 650 km²) study areas according to 'high' (4), medium (3) or low (4) density of encounters. An encounter density surface was computed from the 228 records, and study areas located according to tertiles of the data[16]. Study areas contained multiple villages of different physical and population sizes.

Following a pilot ($n = 63$) and questionnaire revisions, Indonesian enumerators gathered data from a sample of male and female heads-of-households between November 2014 and July 2016[16]. To account for multiple families living in one house, a systematic sample of 10% of families per village was achieved by, starting

at the village-head's house, surveying every fourth house in the settlement. Once a house was identified, the sex of the invited respondent was chosen at random by enumerators selecting one of four coloured counters from their pocket: green (three counters) for male; red (one counter) for female. Survey effort was biased towards men because our tiger-encounter database and subsequent pilot revealed that males experienced most encounters with tigers, and they were more likely to make hunting decisions than women. A total of 2386 people were surveyed and missing data were <1.3% for all items. Ethical approval was granted by the School of Anthropology and Conservation Research Ethics Advisory Group, University of Kent. Free prior informed consent was obtained verbally from all participants.

The questionnaire comprised seven sections (Supplementary Table 3) to examine factors underlying people's preference for the size of the local tiger population including: affect, attitude, norms, beliefs and trust in authority. To understand how emotional responses to tigers influence how many tigers people want to exist locally, we measured affective responses on two semantic scales concerning value and danger (Good to Bad, and Dangerous to Harmless; variables, BadGood and DangHarm) after respondents were shown an image of a tiger. Answers to other questions were given on five-point Likert scales (Strongly agree to Strongly disagree). Prevailing attitudes towards the existence of tigers were captured using two target, action, context and time-specific statements concerning whether tigers should be caught (Kill tiger) or protected (Protect tiger). To investigate the relationship of descriptive (perceptions of what most people do) and injunctive (what most people approve/disapprove of) norms on people's support for tigers, respondents were asked to indicate if they felt that most people like them try to hunt tigers (Descriptive), and if they felt pressure to catch tigers themselves (Injunctive). Beliefs of the costs and benefits of living with tigers were measured across three dimensions of well-being: spiritual (Spirit), physical (Health) and environmental (Env). Two statements measured levels of trust in wildlife management authorities to manage wildlife appropriately (TrustA) and keep people safe from animals (TrustB). Responses to these statements were correlated (Pearson's $r = 0.69$, $p < 0.001$), and so only TrustB was retained. Finally, respondents were presented with four typical conflict scenarios and asked to indicate what intervention they would undertake if they were responsible for population management: the four scenarios represented the categories in our encounter database: a tiger is seen but poses no threat to people (ScenA); seen but poses a threat (ScenB); attacks livestock (ScenC); or attacks a person (ScenD). Responses to these questions grouped into a single dimension via factor analysis (Cronbach's $\alpha = 0.78$; 63.9% variance explained), with factor loadings being greater for attacks (ScenC = 0.93; ScenD = 0.79), than for sightings (ScenB = 0.64; ScenA = 0.38). The Kaiser–Meyer–Olkin (KMO) measure verified the sampling adequacy (KMO = 0.72) and all KMO values for individual variables were >0.6. We therefore used a single scenario variable in subsequent modelling (Scenario), which was the average score across the four responses in the questionnaire. Demographic details including ethnicity and sex were recorded, but not respondent identity. All data were assigned coordinates to the centre of the settlement so that household location was protected.

Prior to analysis, all tolerance variables were scaled so that higher values indicated supporting an increasing tiger population. We used Pearson's correlation coefficients to assess collinearity. Respondents with missing data were excluded from modelling. To examine relationships between respondents' tolerance level and their beliefs and perceptions, we modelled categorical responses using multinomial logistic regression models in the R package nnet[52]. The same outcomes were evident when models were run as ordered logit models, but we report multinomial results since our response variable is best perceived as being categorical. Models were selected based on ΔAICc <2, and parameter and error estimates were derived by model averaging of top model(s)[53]. Variables concerning people's affect, attitude, norms, beliefs and trust in authority, as well as their ethnicity, age, gender and average response to tiger management scenarios, were considered as fixed effects. Models were repeated with and without the ecological measures of encounter risk and covariates.

**Data availability.** Encounter localities, socio-ecological data and the risk and tolerance profiles of villages in the prioritisation framework are available in the repository https://doi.org/10.22024/unikent/01.01.37. The identities of individual villages have been removed from these files to ensure anonymity of respondents. Further information is available from the corresponding author on reasonable request.

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

## Acknowledgements

We thank Ika Agustin, Yulian Anggriawan, Karlina, and Erlinda Kartika for implementing the questionnaire surveys, Darmawan Liswanto for facilitating the study, and the thousands of people in Sumatra who shared their experiences living with tigers. This work was funded by a UK Leverhulme Trust Research Project Grant, and facilitated in Sumatra by Fauna and Flora International Indonesia programme.

## Author contributions

M.J.S., F.A.V.St.J., M.L., N.L.-W., F.M.M. and J.E.M. designed the study; D.J.M. provided data on human–tiger encounters; B.M. and J.E.M. managed the field team; M.J.S. and N.J.D. undertook binomial models; S.C.F. and S.C.L.C. the geographic profiling and F.A.V.St.J. analysed social data. All authors discussed results and edited the manuscript.

## Additional information

**Competing interests:** The authors declare no competing interests.

