## [Peer Review File · Nature Communications]

Reviewers' comments:

Reviewer #1 (Remarks to the Author):

Thank you for the opportunity to review this manuscript. I have provided some comments on the PDF attached to this review.

[Editorial Note: See next page for reviewer comments]

Summary of Comments on
1_reviewer_attachment_1_1523130798_convrt.pdf

Number: 1 Author: Date: 4/7/2018 12:39:06 PM

The last sentence doesn't really flow from the earlier text. Not sure where it comes from and it seems very overgeneralized.

Number: 1 Author: Date: 4/7/2018 12:46:02 PM

The case of Sumatra seems a bit unique, particularly when the religious bits are added in. I wonder if it is possible to walk the reader through how the use case is relevant to other species and contexts? If not, it is hard to extrapolate and generalize.

Number: 2 Author: Date: 4/7/2018 12:48:06 PM

Is there something unique that should be said about coordinating with folks inside a protected versus outside a protected area? The PA system may be a confounding factor in tolerance now? Shouldn't that be addressed somewhere, if not here then perhaps someplace else?

Page: 7

Number: 1 Author: Date: 4/7/2018 12:48:27 PM
Why this date range?

Number: 2 Author: Date: 4/7/2018 3:01:16 PM
I'm not sure I understand what you mean with this last sentence here. I also am not sure I see the connection between negative trends elsewhere and the high numbers/high tolerance situation in Sumatra.

Number: 3 Author: Date: 4/7/2018 3:02:02 PM
Wow. This sounds super cool!

Number: 1 Author: Date: 4/7/2018 3:05:19 PM

This seems to be an over generalization. The protected area has a substantial edge and also there are unique aspects of the socio-ecological system, as you point out, that are influencing assessed risk. I wonder if this statement can simply be revised so as to clarify and not overgeneralize?

Number: 1 Author: Date: 4/7/2018 3:06:23 PM

I'm not clear as to how this is a measure of tolerance for living with tigers. There are a number of established measures for tolerance. Is this one of them? Why not use one of the established measures?

Number: 2 Author: Date: 4/7/2018 3:07:05 PM

Why multinomial logistic? I can't tell from how the manuscript is written if the sample was random?

Number: 1 Author: Date: 4/7/2018 3:08:49 PM

It seems like there may be a conflation b/w religion and ethnicity here? Weren't the reasons delineated earlier in the manuscript associated with religion and not ethnicity? This is a bit confusing for me.

Number: 1 Author: Date: 4/7/2018 3:10:36 PM

Hard to assess if this is considered high, low etc? Are these numbers to be expected? Why or why not?

Number: 2 Author: Date: 4/7/2018 3:11:21 PM

What is your expectation of how tolerance and risk would be displaced given the interventions you are proposing? How do you know they wouldn't just displace or move?

Number: 1 Author: Date: 4/7/2018 3:12:50 PM

This is an overgeneralization that is not equivocated by the data. Certainly this work challenges the assumption, but the results may be the exception to the rule and no

Number: 1 Author: Date: 4/7/2018 3:43:14 PM

It would be interesting to know your thoughts about the types of HWC you measured. There can be a lot of other types of interactions and incorporating them in the study or discussion could be very interesting.

Number: 1 Author: Date: 4/7/2018 3:43:52 PM

What are the implications of using such dated data? Almost a decade old in some instances?

Reviewer #2 [Comments to Authors]

Review of "Tolerating tigers: Do local beliefs offset human-wildlife conflicts?"

The article provides a rare social-ecological analysis and perspective on drivers of human-wildlife conflict. The study's novelty and importance come through addressing several long-hypothesized relationships related to the role that spiritual beliefs play in shaping tolerance for and responses to wildlife in Asian societies. The framework advances our understanding of human-wildlife conflict through a socio-ecological lens and offers a pragmatic way of presenting considerations between different metrics with clear application to conservation and conflict mitigation. Thus, the paper is timely, novel and of relevance for scientific discussions of this topic. Statistical analyses were conducted appropriately and built sequentially into the methods in an interesting and informative way. That said, I offer one comment that should be addressed at least briefly in the methods with regards to the selection of random points.

Major comment:

L396: The location of random points can significantly change the outcome of a model. There is debate over whether random locations in human-carnivore conflict risk models should represent the entire "background" (all possible locations where incidents could occur) or represent a "paired" design (where a certain number of random points are focused around each presence point to better represent the environmental alternative choices in each incident case which were not selected, e.g. Treves et al. 2004). Regardless of this choice (and I would be curious to hear the authors explain their choice), the random points should be limited to areas where conflict incidents are possible. In this case, that means locations where tigers, livestock and people are or could be (areas with a high probability of occupancy, or at least constrained to exclude areas where these species are not). Please explain and justify (in your paper methods) how you narrowed down the random point selection area to represent probable areas of conflict which were not selected by tigers. If you did not, I strongly urge you to consider doing so, or at the very least consider how your results may have been impacted by this choice.

Treves A, Naughton-treves L, Harper EK, Mladenoff DJ, Rose RA, Sickley TA, Wydeven AP. Predicting human-carnivore conflict: a spatial model derived from 25 years of data on wolf predation on livestock. *Conserv Biol* 2004; 18:114–25.

Minor comments:

L205: change semi-colon to comma.

L208-211: This seems intuitive and likely to be how law enforcement would base their decisions without any scientific guidance. Please provide an additional sentence or two of context so the reader understands whether this is new advice, and how it would potentially change how patrols currently protect tigers.

L219: This question implicitly assumes that the participants all are answering based on same understanding of how a change in tiger population would affect them, yet this is not likely the case. For example, one participant might believe that an increase in tiger population would cause more conflict (e.g. attacks on people or livestock) whereas a second participant might believe an increase would not cause a change in conflict. Would it be better (fewer assumptions, more explicit connection between the question and 'tolerance') if the question were linked to an effect more directly? Clearly this methodology cannot be addressed at this stage in the research, but I'm curious to hear the authors' thought on this issue and one decides how to frame questions about 'tolerance'. I also

wonder how the current framing of the question might affect the outcome, or the authors' assumptions about what 'tolerance' is measuring.

L231-232: Please explain what each of these steps (1) including ecological covariates from the risk models to the social covariations, 2) including output probabilities from the ensemble model of risk and 3) including geographic profile measure of risk) would have meant for the model/results. Readers less familiar with these methods would benefit from a brief explanation here as to why you tried these.

Fig 2. In the Figure, please show a word or phrase for the coded covariates (e.g. AttKill, InjNorm) that is informative for the reader (who doesn't know this shorthand). This would be preferable to defining the shorthand in the caption so the figure stands alone without the caption. Also, labeling the neutral and 'pro-conservation' and 'anti-conservation' spaces on the figure would be equally helpful.

L265: Delete comma.

L279: Comma placement should read "those opting for an increase, rather than a reduction, in tigers had higher..."

L307: The term "geographic profile" isn't particularly intuitive. I would recommend changing to "sightings" or "encounters" or "observations".

L352: Millions of "U.S." dollars? Clarify the currency and scale.

L356: Replace semi-colon with comma.

L360-362: This disconnect could also be indicative of an ineffective use (or lack of use) of predator deterrents, and expose a need for authorities to invest resources, training or awareness of such interventions. It may be helpful to reference the discussion of this in disconnected observed. Actual. link your findings to those in Miller et al. 2017, which found that owners who lost livestock for the first time were more receptive to changing their livestock protection measures (via shifting grazing) than owners who had previously lost livestock (e.g. were habituated to the problem).

Miller JRB, Jhala Y V., Jena J, Schmitz OJ. Human perceptions mirror realities of carnivore attack risk for livestock: implications for mitigating human-carnivore conflict. PLoS One 2016; 11:e0162685.

L419: change "burnin" to "burn-in"

Reviewer #3 (Remarks to the Author):

This paper tackles a very topical issue in terms of predicting and mitigating attacks from and against tigers in Asia. The authors also use an interdisciplinary approach, combining ecological and social components to map conflict hotspots, thereby reducing the uncertainties associated with using only one system component. I was also impressed by the number of surveys carried out.

I thought the paper was very well written, and, apart from a few instances, would appeal to a wide audience including practitioners.

There are, however, some key aspects that need to be justified, clarified or expanded upon:

- The case study is focussed on Sumatra, which, as you explain in the paper, is unusual on many counts, including their spiritual beliefs. As such, there is a specificity or a context-dependence for this specific case which makes it difficult to justify your extrapolation to how your approach could be used elsewhere in terms of offsetting hostility towards carnivores. I understand fully how your approach can help predict conflict hotspots and prioritise mitigation efforts, but I am not convinced the same approach would "overcome negative trends elsewhere", simply because of the uniqueness of your case study.

- An interesting result was the validity of the sighting data alone in alerting authorities of future incidents with tigers. I was, however, very unclear over the difference between prioritising intervention using sighting data alone, versus prioritising intervention using information on risk and tolerance. I understand that sighting data may not be available in other areas, but then again, gathering data on attitudes and beliefs is resource-intensive. I would therefore welcome more justification of what the social data on tolerance adds to the understanding of the system in terms of mitigating impacts.

- The issues of tolerance and co-existence (and the links between these two terms) needs to be defined. These are obviously key terms but are used rather loosely in the paper. For example, is co-existence a lack of retaliation killing? Or would we aspire for more, in which case what, and defined by whom?

- Reading through the paper, I thought the use of human-wildlife conflict was not adequately describing the situation, which appeared to be more of an impact rather than a conflict (see Peterson et al., 2010; Redpath et al., 2013). Indeed, many of the mitigation strategies alluded to in the paper (lines 148-149) would tally more with the definition of an impact.

- The Tiger Protection and Conservation Units are a very interesting element to the case study. Are they locally led or is this an externally driven initiative? Is there is local ownership and trust towards the Units? You mention trust in line 278 onwards but it is unclear what is meant by the "management authorities". These issues are often key in mitigating impacts.

- You mention a spike in poaching since 2013 - it would be interesting to understand more about this, in terms of who is responsible (are they part of or from other communities?), how does the poaching link with the changes in the local communities, and how these changes affect the parameters that you explored.

- The predictive performance of the spatial models for encounter risk was only fair – particularly given the number of observations - suggesting that environmental variables used in the model were not contributing significantly to predictive performance.

- The use of a multinomial model for a Likert scale is questionable particularly as the authors state they also ran ordered logit models, which would be far more appropriate in this case.

Reviewer #4 (Remarks to the Author):

Review of: Tolerating tigers: Do local beliefs offset human-wildlife conflicts?

Manuscript ID: NCOMMS-18-06811

Recommendation

Accept, with minor revisions.

Summary

This is an interesting and well-written paper. The study involves novel techniques and potentially useful findings.

Important Issues

The section on geographic profiling (pp. 9, 20) focuses almost exclusively on details of the Dirichlet Process Mixture (DPM) model; there is no discussion of its underlying assumptions, what the model is doing, and, most importantly, its relationship to tiger movement and hunting patterns. There should be an explanation of why geographic profiling is superior to a simple kernel density estimation. A KDE is based on the sightings, attacks, and other locations being analyzed, while a geoprofile shows the most probable area of origin for these sites; there needs to be some justification for the decision to use the latter (see Rossmo, 2000). Adding a short discussion of the size of tiger hunting ranges, probability of different tigers overlapping in the Kerinci Seblat, and other relevant animal behaviors would better connect the model to reality.

Minor Issues

The encounter probability scale in Figure 1(c) is incorrect (p. 8); probabilities range from 0 to 1.

"We surveyed 2,386 people across Kerinci Seblat, who were mostly male (73.9%), middle-aged (mean age 43.8 years; SE = 0.26) ..." (p. 10). This is confusing as written. Were most of the surveyed people middle-aged, as the sentence implies, or was only the average in this age group? If the former, why were younger and older people not surveyed? Any age bias could have a significant impact on attitudes and norms. If the latter, the sentence needs to be correctly written.

"Prior to 2013, the removal of at least 15 tigers (65% of all reported) may have been avoided if these villages had been prioritised earlier" (p. 15). If the data from these 15 tigers were removed from the analysis, would the results have been similar (i.e., would the loss of this information have influenced the model's subsequent prediction accuracy)?

Typographical and Grammatical Errors

"a small amount of sightings" should be "a small number of sightings" (p. 15).

"Tiger occupancy was based on surveys ... which was then spatially interpolated to non-survey areas" should be "Tiger occupancy was based on surveys ... which were then spatially interpolated to non-survey areas" (p. 19).

"physical & and population sizes" should be "physical and population sizes" (p. 21).

"people an tigers" should be "people and tigers" (p. S6).

References

Rossmo, D. K. (2000). Geographic profiling. Boca Raton, FL: CRC Press.

NCOMMS-18-06811 (Tolerating tigers: Do local beliefs offset human-wildlife conflicts?)

RESPONSE TO REVIEWERS:

Reviewer #1 (Remarks to the Author):

Thank you for the opportunity to review this manuscript. I have provided some comments on the PDF attached to this review.

(comments below are therefore copied from the pdf version supplied)

Page 2; line 40: The last sentence doesn't really flow from the earlier text. Not sure where it comes from and it seems very overgeneralized.

We have made some minor changes to the abstract to help emphasize that the last sentence refers to the integrated findings from the risk and tolerance models. With a narrow word limit for the abstract we welcome advice on how to elaborate this further, but hope that the reviewer's comment concerning generalisation arose before (s)he had fully read the manuscript.

Page 6; line 130: The case of Sumatra seems a bit unique, particularly when the religious bits are added in. I wonder if it is possible to walk the reader through how the use case is relevant to other species and contexts? If not, it is hard to extrapolate and generalize.

Our key point is that the analytical framework we use could be locally adapted to other human wildlife conflict systems in which the interplay between ecological risk and social tolerance is important (i.e. most conflict systems to some degree). In light of this comment we have made a minor edit to penultimate paragraph in the introduction as follows:

Line 158 "In the continued absence of encounter data it is therefore important to prioritise mitigation efforts to reduce casualties of livestock, people and tigers, and learn lessons from this fascinating system which could be applied to other socio-ecological contexts."

Our main change has been in the closing paragraph of the Discussion in which we highlight the broader relevance of the study:

Line 390 '...is still reason for optimism, and serves as an incentive to expand our framework to other priority human-wildlife conflict systems elsewhere. Other conflict species reported from the region include elephants, orang-utans and bears. Although the spiritual connections with wildlife reported from Sumatra are somewhat unique, cultural tolerance is known to enable greater persistence of carnivore species in regions as diverse as India⁴⁴ and Ethiopia⁴⁵. Where such beliefs are not widely held, or are outweighed by fear, retaliatory killings increase, as found for jaguars in Brazil⁴². Therefore, the socio-ecological interplay between risk and tolerance permeates many human wildlife conflict situations, and so locally adapting our framework to these contexts could help avoid further losses to people and some of the world's most endangered species.

Page 6; line 141: Is there something unique that should be said about coordinating with folks inside a protected versus outside a protected area? The PA system may be a confounding factor in tolerance now? Shouldn't that be addressed somewhere, if not here then perhaps someplace else?

We're confused about what the reviewer is referring to or requesting here. The Units have benefited from coordination with PA patrol teams, but the data utilised in our study came from reports to the Units from the public, and so are broadly representative of other reporting systems set up in other human-wildlife conflict systems.

Page 7; line 144: Why this date range?

It is implicit in this section that the Units were operational from the year 2000, and that reporting ceased in early 2014. Therefore, data were only available from 2000 to 2013.

Page 7; line 156: I'm not sure I understand what you mean with this last sentence here. I also am not sure I see the connection between negative trends elsewhere and the high numbers/high tolerance situation in Sumatra.

We have now revised this sentence so that the reviewer's query should no longer apply.

Page 7: line 162: Wow. This sounds super cool!

Glad you like it!

Page 9; line 209: This seems to be an over generalization. The protected area has a substantial edge and also there are unique aspects of the socio-ecological system, as you point out, that are influencing assessed risk. I wonder if this statement can simply be revised so as to clarify and not overgeneralize?

We made a minor revision to the sentence (to 'help alert response units of potential future incidents') to soften the statement as requested by the reviewer.

Page 10; line 219: I'm not clear as to how this is a measure of tolerance for living with tigers. There are a number of established measures for tolerance. Is this one of them? Why not use one of the established measures?

Our approach to measuring tolerance is grounded in the work of authors such as and Riley and Decker (2000) who proposed the concept of wildlife acceptance capacity, which entails measuring respondent's preferences for species population levels. We have inserted an extra row in the top of Supplementary Table 3 so that the statement used to measure this variable is now available to readers. Further, we have revised the sentence referred to by the reviewer to clarify the grounding of this measure in the wildlife conflict literature:

Line 222 'We measured tolerance as the capacity for people to accept wildlife (sensu^{17,40}) by asking respondents whether they would prefer the tiger population to be reduced/eradicated, stay the same, or increase.'

Page 10; line 230: Why multinomial logistic? I can't tell from how the manuscript is written if the sample was random?

As stated in the Methods section of the manuscript, we used multinomial, as opposed to ordinal logistic regression, because our response variable is best viewed as a categorical variable (1. Eradicated/reduced 2. Kept the same 3. Increased). To further clarify why this variable is treated as categorical rather than continuous, we have added a footnote to the revised Supplementary Table 3 which describes the variables investigated:

'Four-point scale: 'Completely eradicate', 'Reduce in number', 'Kept the same in number'. 'Don't know' was also permitted as a response. Prior to analysis, 'Completely eradicate' and 'Reduce in number' were collapse into one category with the variable treated as categorical thereafter.'

Note that we re-ran these models using ordered logit models, and had the same outcome. However, we maintain our use of multinomial logistic for the reasons stated.

Page 13; line 263: It seems like there may be a conflation b/w religion and ethnicity here? Weren't the reasons delineated earlier in the manuscript associated with religion and not ethnicity? This is a bit confusing for me.

Sorry for the confusion caused here. To address this comment we have edited an earlier sentence (originally line 125) which misleadingly implied that we investigate how differences in religious beliefs may relate to differences in tolerance for tigers. We do not do this as all respondents followed a single religion - Islam. Rather, the beliefs referred to are thought to be inherent to certain ethnic groups in the area (although our analyses later indicate this may not actually be the case)

Line 129 'The spiritual belief systems of some ethnic groups, coupled with on-going ecological monitoring of tigers and their encounters with people...'

Page 15; line 302: Hard to assess if this is considered high, low etc? Are these numbers to be expected? Why or why not?

This comment refers to the statement "At least 40 attacks of livestock and people were reported from the high priority settlements in the 13 years of records; 54% of all attacks in our study villages. Prior to 2013, the removal of at least 15 tigers (65% of all reported) may have been avoided if these villages had been prioritised earlier".

Here we have defined what proportion of the dataset are represented in these villages (54% of all attacks on livestock or people; and 65% of all tigers removed), and so it should be implicit that this is a significant finding. Nonetheless, we have added a sentence to point this out given the low population size of tigers and sheer size of the study area:

Line 310 "Prior to 2013, the removal of at least 15 tigers (65% of all reported) may have been avoided if these villages had been prioritised earlier; a highly significant number considering the rarity of tigers and the size of the Kerinci Seblat landscape."

Page 15; line 308: What is your expectation of how tolerance and risk would displace given the interventions you are proposing? How do you know they wouldn't just displace or move?

If we understand this comment correctly the reviewer is asking whether interventions could end up displacing the conflict elsewhere? We believe this to be highly unlikely. Tigers maintain well defined home ranges, and so resident tigers are unlikely to be displaced beyond their range (ca. 250-300 km²). If the problem tiger is a transient individual searching to establish its own range then it is conceivable that preventing it from attacking livestock in one area could potentially result in it moving to another area. However, transients are more likely to be directing resources to establishing a core home range to start breeding than to venture outside of their core habitat searching for prey.

While an interesting point we feel that elaborating on this in the text could detract from the core messages of the manuscript, and so we have not expanded the Discussion. However, we can accommodate the change if the reviewer or editor believes it to be central to the study.

Page 17; line 336: This is an overgeneralization that is not equivocated by the data. Certainly this work challenges the assumption, but the results may be the exception to the rule and no *[sic – comment end here in the pdf]*

It is unclear what the reviewer is specifically referring to here as their comment ends abruptly in the pdf. The sentence preceding and proceeding their comment reads:

'Rather, our findings indicate that tolerance towards tigers in Kerinci Seblat at least, is driven by a number of complex factors that include spirituality, as well as risk of actual attack. Notably, respondents in our landscape were more likely to support an increase in tiger populations if they held beliefs concerning the importance of the animal for spiritual, environmental and physical wellbeing, but this inclination appeared broadly prevalent amongst respondents and independent of their ethnicity (Table 2).'

Both of these statements are supported by our tolerance models, and qualified by us referring to the Kerinci Seblat system and our study findings. If the reviewer can point us towards the part they believe to be overgeneralized we will gladly make the change. At the moment we cannot respond in full because of the way they have commented on the pdf.

Page 18; line 369: It would be interesting to know your thoughts about the types of HWC you measured. There can be a lot of other types of interactions and incorporating them in the study or discussion could be very interesting.

As described in the core text, the information we used came from an archive of reports from the Tiger Units over the 13 years in operation. These took the form of (faded) facsimiles, photocopies, word processing documents, and emails from up to 6 different teams working over a large area and in three languages. Compiling and categorising this information was no easy task. We therefore chose the four categories – sightings > livestock attacks > people attacks > tigers hunted – in part to reflect how HWC incidents are reported elsewhere in the literature, and how they are conceptualised in Sumatra, but also because of limitations with the information available. For example, we had initially intended to quantify the numbers of livestock attacked, because in some instances loss of life was substantial (> 10 goats plus several guard dogs in some incidents for example). However, we sometimes found inconsistent numbers reported by various Unit personnel (e.g. first response documented 8 goats, but on-site conflict resolution staff reported more), and so decided to restrict our analyses to incidents (i.e. occurrence of an attack) rather than frequencies (e.g. number of livestock killed). That said, although delving deeper into the encounter data was desirable, there are clear patterns as analyses, and we demonstrate convincingly how even sparse information on human-wildlife encounters can be useful for prioritising intervention.

It sounds like the reviewer is convinced of this, but sought some clarification in our response. We hope they remain convinced that they concur we have done the best job with the data available.

Page 19; line 383: What are the implications of using such dated data? Almost a decade old in some instances?

The occupancy data were drawn from surveys undertaken between 2007 and 2009, which is approximately mid-way through the study timeframe (2000-2013). While these data are dated to some degree, they are all that is currently available for the whole landscape, and they are representative of the 13 year investigation. We already refer to this in the Discussion, and make a suggestion for improvement where data exist:

Line 379: 'For example, tiger occupancy data were informative to our encounter models, but were based on past survey data, after which densities are thought to have declined 27. Incorporating dynamic variables into the ecological models as survey coverage is expanded could therefore improve our predictions.'

Reviewer #2 [Comments to Authors]

Review of “Tolerating tigers: Do local beliefs offset human-wildlife conflicts?”

The article provides a rare social-ecological analysis and perspective on drivers of human-wildlife conflict. The study's novelty and importance come through addressing several long-hypothesized relationships related to the role that spiritual beliefs play in shaping tolerance for and responses to wildlife in Asian societies. The framework advances our understanding of human-wildlife conflict through a socio-ecological lens and offers a pragmatic way of presenting considerations between different metrics with clear application to conservation and conflict mitigation. Thus, the paper is timely, novel and of relevance for scientific discussions of this topic. Statistical analyses were conducted appropriately and built sequentially into the methods in an interesting and informative way. That said, I offer one comment that should be addressed at least briefly in the methods with regards to the selection of random points.

Major comment:

L396: The location of random points can significantly change the outcome of a model. There is debate over whether random locations in human-carnivore conflict risk models should represent the entire “background” (all possible locations where incidents could occur) or represent a “paired” design (where a certain number of random points are focused around each presence point to better represent the environmental alternative choices in each incident case which were not selected, e.g. Treves et al. 2004). Regardless of this choice (and I would be curious to hear the authors explain their choice), the random points should be limited to areas where conflict incidents are possible. In this case, that means locations where tigers, livestock and people are or could be (areas with a high probability of occupancy, or at least constrained to exclude areas where these species are not). Please explain and justify (in your paper methods) how you narrowed down the random point selection area to represent probable areas of conflict which were not selected by tigers. If you did not, I strongly urge you to consider doing so, or at the very least consider how your results may have been impacted by this choice.

Treves A, Naughton-treves L, Harper EK, Mladenoff DJ, Rose RA, Sickley TA, Wydeven AP. Predicting human-carnivore conflict: a spatial model derived from 25 years of data on wolf predation on livestock. *Conserv Biol* 2004; 18:114–25.

Our study landscape was defined as all village administrative areas that were within 5km of the core forest area (including Kerinci Seblat, and surrounding forest reserves), since tiger encounters had been reported within this area over the 13 year period. Therefore, given these encounter patterns it is feasible that future incidents could take place within this area. In addition, although the 2007-2009 tiger surveys indicated that tiger occupancy was much greater in a core area surrounding Kerinci Seblat, with the exception of highly urban areas, at no point across the landscape was occupancy predicted to be zero (i.e. absent). Previous and ongoing track and sign survey data indicate that tigers are detected across the whole study region. The random points were therefore drawn from the same region as the source encounter data.

The reviewer raises a good point that this rationale should be improved in the text. In our revised manuscript we have therefore started the Methods section by defining the study area:

Line 403: ‘We defined our study landscape by a 5km buffer around the Kerinci Seblat national park and adjacent forest reserves, within which the vast majority of encounter records were reported.’

We then elaborate on the random point selection in the next section, citing the manuscript suggested by the reviewer:

Line 423: ‘All 228 incidents were assigned a value of “1”, and 10,000 localities were randomly drawn from the background and assigned “0”. Although there is some debate regarding whether this background selection should be restricted⁴⁷, we drew our pseudo-absence data from the full farmland extent since the encounter database indicated that incidents were possible over the whole landscape over the 13 years of records.’

Minor comments:

L205: change semi-colon to comma.

Corrected as requested

L208-211: This seems intuitive and likely to be how law enforcement would base their decisions without any scientific guidance. Please provide an additional sentence or two of context so the reader understands whether this is new advice, and how it would potentially change how patrols currently protect tigers.

The important point here is that sighting data are more informative in some areas than others. We have now revised this sentence as follows:

Line 211: 'Therefore, sighting data from a few specific areas are more informative to help alert response units of potential future incidents before they escalate to injury or loss of life.'

L219: This question implicitly assumes that the participants all are answering based on same understanding of how a change in tiger population would affect them, yet this is not likely the case. For example, one participant might believe that an increase in tiger population would cause more conflict (e.g. attacks on people or livestock) whereas a second participant might believe an increase would not cause a change in conflict. Would it be better (fewer assumptions, more explicit connection between the question and 'tolerance') if the question were linked to an effect more directly? Clearly this methodology cannot be addressed at this stage in the research, but I'm curious to hear the authors' thought on this issue and one decides how to frame questions about 'tolerance'. I also wonder how the current framing of the question might affect the outcome, or the authors' assumptions about what 'tolerance' is measuring.

Thank you for raising this interesting view point. It is something that we will bear in mind when designing future studies. We have added new text to the discussion to capture this point:

LINE 357 'Measures of tolerance may better reflect people's actual ability to coexist with a species if linked explicitly to an effect. For example, the probability of increased human-tiger interaction given a specified population increase. However, data to specify such statements are lacking in most systems'.

L231-232: Please explain what each of these steps (1) including ecological covariates from the risk models to the social covariations, 2) including output probabilities from the ensemble model of risk and 3) including geographic profile measure of risk) would have meant for the model/results. Readers less familiar with these methods would benefit from a brief explanation here as to why you tried these.

Thanks for pointing this out. We have now added a sentence preceding this statement to describe what our expected outcome would be if ecological and social factors were important:

Line 231: "A multinomial logistic regression model was used to examine the relationships between these social-psychological variables and tolerance amongst respondents (Table 1). This model was then repeated with risk covariates included to determine the extent to which tolerance was defined by both ecological and social factors (i.e. with the expectation that this would improve model performance)."

Fig 2. In the Figure, please show a word or phrase for the coded covariates (e.g. AttKill, InjNorm) that is informative for the reader (who doesn't know this shorthand). This would be preferable to defining the shorthand in the caption so the figure stands alone without the caption. Also, labeling the neutral and 'pro-

conservation' and 'anti-conservation' spaces on the figure would be equally helpful.

We have made these changes as requested. We are not keen to include both 'pro-' and 'anti-conservation' as the questions were not posed in these terms. Instead we have used an arrow alongside the axis to indicate the direction of pro-conservation scores.

L265: Delete comma.

Done

L279: Comma placement should read "those opting for an increase, rather than a reduction, in tigers had higher..."

Corrected

L307: The term "geographic profile" isn't particularly intuitive. I would recommend changing to "sightings" or "encounters" or "observations".

We believe it is important to consistently distinguish the two mapped measures of risk in the text, and refer to them by the methodologies used. Therefore we have corrected this text to further emphasize that the geographic profile output is the second measure of risk:

Line 314: "If prioritisations were made based on the geographic profile measure of risk coupled with tolerance data."

L352: Millions of "U.S." dollars? Clarify the currency and scale.

Revised to U.S. dollars as suggested.

L356: Replace semi-colon with comma.

Done

L360-362: This disconnect could also be indicative of an ineffective use (or lack of use) of predator deterrents, and expose a need for authorities to invest resources, training or awareness of such interventions. It may be helpful to reference the discussion of this in disconnected observed. Actual. link your findings to those in Miller et al. 2017, which found that owners who lost livestock for the first time were more receptive to changing their livestock protection measures (via shifting grazing) than owners who had previously lost livestock (e.g. were habituated to the problem).

Miller JRB, Jhala Y V., Jena J, Schmitz OJ. Human perceptions mirror realities of carnivore attack risk for livestock: implications for mitigating human-carnivore conflict. PLoS One 2016; 11:e0162685.

Thanks for this insight. However, at this point in the manuscript we're not sure it is helpful as the situation in Kanha Tiger Reserve (where the Miller research was based) is very different to that in Kerinci Seblat. In Kanha the number of livestock kept by villagers is much higher than in Kerinci, where crop cultivation rather than livestock keeping dominates; those who keep livestock do so in low numbers compared to around Kanha. Further, the density of large carnivores in Kahna is 3 times higher (tiger only) and 8 times higher (tiger/leopard) than Kerinci. Thus the capacity for villagers to adapt their livestock is much greater in Kahna since much more of their livelihood is devoted to this

practice. We have therefore refrained from including this material given the differences between the two systems.

L419: change “burnin” to “burn-in”

Done

Reviewer #3 (Remarks to the Author):

This paper tackles a very topical issue in terms of predicting and mitigating attacks from and against tigers in Asia. The authors also use an interdisciplinary approach, combining ecological and social components to map conflict hotspots, thereby reducing the uncertainties associated with using only one system component. I was also impressed by the number of surveys carried out.

I thought the paper was very well written, and, apart from a few instances, would appeal to a wide audience including practitioners.

There are, however, some key aspects that need to be justified, clarified or expanded upon:

- The case study is focussed on Sumatra, which, as you explain in the paper, is unusual on many counts, including their spiritual beliefs. As such, there is a specificity or a context-dependence for this specific case which makes it difficult to justify your extrapolation to how your approach could be used elsewhere in terms of offsetting hostility towards carnivores. I understand fully how your approach can help predict conflict hotspots and prioritise mitigation efforts, but I am not convinced the same approach would “overcome negative trends elsewhere”, simply because of the uniqueness of your case study.

We have now addressed this comment also in our response to Reviewer 1. First we made a minor edit to penultimate paragraph in the introduction as follows:

Line 157 “In the continued absence of encounter data it is therefore important to prioritise mitigation efforts to reduce casualties of livestock, people and tigers, and learn lessons from this fascinating system which could be applied to other human-wildlife systems.”

Our main change has been in the closing paragraph of the Discussion in which we highlight the broader relevance of the study:

Line 390 ‘...is still reason for optimism, and serves as an incentive to expand our framework to other priority conflict systems. Other conflict species reported from the region include elephants, orang-utans and bears. Although the spiritual connections with wildlife reported from Sumatra are somewhat unique, cultural tolerance is known to enable greater persistence of carnivore species in regions as diverse as India⁴⁴ and Ethiopia⁴⁵. Where such beliefs are not widely held, or are outweighed by fear, retaliatory killings increase, as found for jaguars in Brazil⁴². Therefore, the socio-ecological interplay between risk and tolerance permeates many human wildlife conflict situations, and so locally adapting our framework to these contexts could help avoid further losses to people and some of the world’s most endangered species.

- An interesting result was the validity of the sighting data alone in alerting authorities of future incidents with tigers. I was, however, very unclear over the difference between prioritising intervention using sighting data alone, versus prioritising intervention using information on risk and tolerance. I understand that sighting data may not be available in other areas, but then again, gathering data on attitudes and beliefs is resource-intensive. I would therefore welcome more justification of what the social data on tolerance adds to the understanding of the system in terms of mitigating impacts.

The reviewer is referring to the re-running of both ensemble and geographic profiling models using only sighting encounter data, rather than the full set of encounter localities; the rationale being to

see whether reliable predictions could be made on the whereabouts of subsequent attacks using early warning data (i.e. sightings). To clarify, in the prioritisation presented in Figure 3 we used the risk measures (i.e. ensemble or geoprofile model outputs) based on the full encounter data. As the model outputs based on full encounter data or sightings-only were near-identical, the ranking of villages as high to low priority is the same regardless of which version of each risk model is used. We have now clarified this in the figure legend so it is clearer to the reader:

Line 322 “Fig.3. Prioritisation of 75 survey villages into high (magenta = above median risk; below median tolerance), medium (beige = above median risk; above median tolerance) and low (green = below median risk) priorities for human-tiger mitigation efforts. (a) using an ensemble of binomial risk models, or (b) using a geographic profile. The estimates of encounter risk were based on models utilizing all 228 human-tiger encounters; alternative estimates from models using sightings data only provided the same result. Point size is weighted by the number of attacks (on livestock or people) reported in each village.”

The reviewer makes a second query concerning what the social data on tolerance adds to the understanding of the system in terms of mitigating impacts. To demonstrate the relative value of the risk and tolerance data we now elaborate on the number of villages prioritised by each feature. We have added a table (S4) to the Supplementary Information that summarises the number and proportion of villages in each section of each plot, which makes it easier for the reader to see where the numbers mentioned in the main text are sourced. We also add a sentence to the first paragraph of the Discussion to elaborate on this:

Line 337 ‘Social models of tolerance that included ecological information on risk were up to 32 times better than using social predictors alone. Furthermore, combining information on tolerance to prioritisations based on risk allows villages to be ranked according to their likelihood of retaliation, allowing valuable mitigation resources to be invested where they are needed most’.

- The issues of tolerance and co-existence (and the links between these two terms) needs to be defined. These are obviously key terms but are used rather loosely in the paper. For example, is co-existence a lack of retaliation killing? Or would we aspire for more, in which case what, and defined by whom?

We have added a definition of tolerance:

Line 71 ‘Tolerance is a passive concept requiring no action, whereas intolerance may be expressed through actions including killing of individual animals or expressing opposition towards existing or increasing population levels^{17,18}’.

We have also made several minor edits in the introduction in order to clarify our use of the terms tolerance and coexistence.

- Reading through the paper, I thought the use of human-wildlife conflict was not adequately describing the situation, which appeared to be more of an impact rather than a conflict (see Peterson et al., 2010; Redpath et al., 2013). Indeed, many of the mitigation strategies alluded to in the paper (lines 148-149) would tally more with the definition of an impact.

We thank the reviewer for this input which we address firstly by making changes throughout the manuscript. For example:

Line 49 the terms conflict has been replaced with ‘greater contact’

Line 51 we now support our use of the term conservation conflict with Redpath et al., 2013

Line 82 we revise our use of HWC to ‘when designing holistic interventions to resolve conservation conflicts ...’

Line 331 Again, we now support our use of the term conservation conflict with a reference (Redpath et al 2013)

Line 364, 370, 375 the term 'human wildlife conflict systems' has been replaced with 'social-ecological systems'

- The Tiger Protection and Conservation Units are a very interesting element to the case study. Are they locally led or is this an externally driven initiative? Is there local ownership and trust towards the Units? You mention trust in line 278 onwards but it is unclear what is meant by the "management authorities". These issues are often key in mitigating impacts.

The Units are run by the national park authority and contain national park staff, supported by community rangers. They therefore include input from both external (government) actors, as well as from local stakeholders. More broadly the authorities referred to here also include Indonesia's Nature Conservation Agency (BKSDA), which is responsible for nature resources and wildlife management in the country. We have added this information in parentheses to clarify this ambiguity:

Line 284 "People's trust in management authorities (national park staff and Indonesia's Nature Conservation Agency) was inconsistently related to tolerance (Table 2):..."

- You mention a spike in poaching since 2013 - it would be interesting to understand more about this, in terms of who is responsible (are they part of or from other communities?), how does the poaching link with the changes in the local communities, and how these changes affect the parameters that you explored.

It is difficult to say exactly who is doing the poaching. There are poaching communities around Kerinci Seblat known to the Tiger Protection and Conservation Units, as well as known outsiders that exploit the area. We expect both poacher types are responding to the increased demand created for tiger skins over time, as described in Linkie et al. 2018. As there was no discernible change in policy, corruption, public concern, or conservation funding during the study period (see Linkie et al. 2018), we do not believe that entire new poaching communities are being created. A more recent reduction in the price of tiger pelts is correlated with reduced poaching rates, but it is too early for us to know for sure whether this trend will continue.

We have elaborated on the poaching context in the Discussion section:

Line 372 'As there was no discernible change in policy, corruption, public concern, or conservation funding for tigers during the study period³⁵, poaching trends are unlikely linked to changes in local communities. Indeed poaching rates are known to fluctuate, and are positively correlated with tiger skin prices and the wider condition of the Indonesian economy³⁵.'

- The predictive performance of the spatial models for encounter risk was only fair – particularly given the number of observations - suggesting that environmental variables used in the model were not contributing significantly to predictive performance.

The measure of predictive performance used, AUC, ranges from '0' to '1' with values >0.5 denoting models considered to discriminate better than chance, and 1 being perfect prediction. Thus our ensemble models (consistently >0.75) make valid predictions. Although there is some debate in the literature about what constitutes "good" or "fair" AUC, arbitrary cut-offs of 0.70 are frequently applied in the conservation planning literature. Strong models are often reported with much higher AUC values, but as with other discrimination metrics, AUC is sensitive to the number of samples, the modelling extent, as well as (importantly) the algorithm used. It is important to note that the AUC values for our consensus models depict the weighted average across 3-5 algorithms. Thus the predictive performance reflects the contributions from algorithms that can themselves be ranked

from “fair” to “good” (0.72-0.85). In this regard, we found that the best performing algorithm was consistently random forests (AUC=0.85-0.86), and although we were tempted to forgo the other algorithms in favour of this approach, we chose not to rely on any single model, and instead incorporated information from several. As there has been a move in the modelling literature away from applying single-algorithm we are convinced this is the appropriate approach to take.

We accept the point that predictive performance could be better. We already devoted text to this in the Discussion, but have elaborated on this further in our revised manuscript:

Line 377 “The finding that substantial numbers of incidents took place in villages deemed low priority under our framework indicates that factors other than risk or tolerance are at play in Kerinci Seblat, and/or additional variables might improve our predictions. For example, tiger occupancy data were informative to our ensemble encounter models, but were based on past survey data, after which densities are thought to have declined²⁷. Incorporating dynamic variables into the ecological models as survey coverage is expanded could therefore improve our predictions. Similarly, there is little information on the availability of prey or accessible livestock, which might also influence tiger encounters^{8,9}. While improved ecological information could help strengthen predictive performance of encounter models, many human-wildlife conflict systems are lacking such information, and so a move to geographic profiling methods could be particularly useful to delineate risk. Moreover, the fact that, despite these shortcomings, substantial numbers of attacks and tiger casualties could still be averted by prioritising responses or intervening proactively (e.g. via tiger-proof livestock enclosures, compensation payments, and de-snaring patrols) is still reason for optimism, and serves as an incentive to expand our framework to other priority human-wildlife conflict systems elsewhere”.

- The use of a multinomial model for a Likert scale is questionable particularly as the authors state they also ran ordered logit models, which would be far more appropriate in this case.

As stated in the Methods section, we used multinomial, as opposed to ordinal logistic regression, because our response variable is best viewed as a categorical variable since two categories were collapsed to one (1. Eradicated/reduced 2. Kept the same 3. Increased). To further clarify why this variable is treated as categorical, rather than continuous, we have added a footnote to the revised Supplementary Table 3 which now incorporates this variable:

‘Four-point scale: ‘Completely eradicate’, ‘Reduce in number’, ‘Kept the same in number’. ‘Don't know’ was also permitted as a response. Prior to analysis, ‘Completely eradicate’ and ‘Reduce in number’ were collapse into one category with the variable treated as categorical thereafter.’

Reviewer #4 (Remarks to the Author):

Recommendation

Accept, with minor revisions.

Summary

This is an interesting and well-written paper. The study involves novel techniques and potentially useful findings.

Important Issues

The section on geographic profiling (pp. 9, 20) focuses almost exclusively on details of the Dirichlet Process Mixture (DPM) model; there is no discussion of its underlying assumptions, what the model is doing, and, most importantly, its relationship to tiger movement and hunting patterns. There should be an explanation of why geographic profiling is superior to a simple kernel density estimation. A KDE is based on the sightings, attacks, and other locations being analyzed, while a geoprofile shows the most probable area of

origin for these sites; there needs to be some justification for the decision to use the latter (see Rossmo, 2000). Adding a short discussion of the size of tiger hunting ranges, probability of different tigers overlapping in the Kerinci Seblat, and other relevant animal behaviors would better connect the model to reality.

As the reviewer asserts, GP models are used to estimate likely sources of incidents, rather than clusters of incidents per se. The technique typically outperforms other widely used spatial statistics, including the simple kernel density approach, and the advantage of GP further improves as the number of possible sources also increases – see Stevenson et al (2012) for example. Key desirable attributes here are that the GP model can run with very few data points (e.g. Rossmo (2000) demonstrated reliable geoprofiles using just 5 data points), and that no further parameters such as habitat choice, dispersal distances or land use, are needed. Therefore, while interesting, including a discussion of the ecological attributes of tigers and people is not informative in this context, because these parameters do not inform the actual model. We recognise that our original description of geoprofiling missed these key descriptions and so we have expanded on this information with a description of in the methods section. Hopefully this is now clearer to the reader why GP is so useful in this context.

Line 439 'Unlike the five algorithms used in our ensemble modelling, geographic profiling is based solely on spatial information, and does not require parameters from other spatial layers or the study system to make predictions. The technique is used to estimate probable sources of spatial data (in this case human-tiger encounters), and is particularly robust when very few data points are used, or when there are an unknown number of sources (e.g. multiple tiger territories, or hunter groups), far outcompeting other spatial statistics across a wide range of contexts⁵¹. Geographic profile models share a distance decay feature, which specifies limitations to travel away from key anchor points (e.g. a tiger territories; livestock areas).'

- Rossmo, D. K. (2000). *Geographic profiling*. Boca Raton, FL: CRC Press.
- Stevenson MD, Rossmo DK, Knell RJ, Le Comber SC. 2012. *Geographic profiling as a novel spatial tool for targeting the control of invasive species*. *Ecography* 35:704-715.

Minor Issues

The encounter probability scale in Figure 1(c) is incorrect (p. 8); probabilities range from 0 to 1.

The reviewer is mistaken: from the geoprofile technique implemented the probabilities should sum to 1 across the entire surface; they cannot reach 1 in any single pixel. As pixel values are very low they are multiplied by 10000 for Figure 1c. This is already indicated in the legend. Note that for the ensemble model (Fig. 1b) probabilities range from 0-1 for each pixel as the reviewer describes.

“We surveyed 2,386 people across Kerinci Seblat, who were mostly male (73.9%), middle-aged (mean age 43.8 years; SE = 0.26) ...” (p. 10). This is confusing as written. Were most of the surveyed people middle-aged, as the sentence implies, or was only the average in this age group? If the former, why were younger and older people not surveyed? Any age bias could have a significant impact on attitudes and norms. If the latter, the sentence needs to be correctly written.

Sorry for the confusion, we were referring to mean age of respondents. We have edited the manuscript text accordingly:

Line 217 “We surveyed 2,386 people across Kerinci Seblat. Most were male (73.9%), and the mean age was 43.8 years (SE = 0.26). Respondents self-identified as Minangkabau (45.4%), Melayu (32.5%), Javanese (7.1%), Rejang (6.5%), Kerincinese (2.9%) or other ethnic group (5.5%)¹⁷”.

“Prior to 2013, the removal of at least 15 tigers (65% of all reported) may have been avoided if these villages had been prioritised earlier” (p. 15). If the data from these 15 tigers were removed from the analysis, would the results have been similar (i.e., would the loss of this information have influenced the model’s subsequent prediction accuracy)?

As part of the original manuscript we re-ran both the ensemble model and geographic profiling analyses using only sighting encounter data. The rationale for this was to see whether reliable predictions could be made using early warning data (i.e. sightings) on the whereabouts of subsequent attacks. As both sightings-only models produced a near-identical outcome (described in lines 197-200 and 205-208, and also presented in Supplementary Figure 3 and 4) we should expect an alternative model based on an intermediate number of attack localities to produce a similar outcome too.

Nevertheless, to allay the Reviewer’s query we re-ran both risk models with the prioritised tiger killings removed, and again produced similar results, thus demonstrating the resilience of the models to source data. In the outputs below it is clear that the geographic patterns of prediction for both the ensemble approach and geographic profiling are similar between the two sets of source data. For the ensemble model we have thresholded the probability of presence using the true skills statistic to make this clearer (note this is not possible for geoprofiling at present, but should be clear from the maps).

We’re not convinced there is value to bring this 3rd analysis into the manuscript given we already present a model based on far fewer locality points. However, if the reviewer or editor believe this is central to the narrative then we could present it in the Supplementary Information. Note, in our revised manuscript we added a figure to the SI to present the two variants of the ensemble modelling, so it is clearer to the reader that we undertook models with the full set of locations and the reduced set of sightings only.

a) Full ensemble model based on all encounters
(AUC = 0.783)

b) Alternate model based on all encounters except
the tiger killings prioritised in a.
(AUC = 0.786)

c) Full geographic profile based on all encounters

d) Alternate profile based on all encounters
except the tiger killings prioritised in a.

Typographical and Grammatical Errors

“a small amount of sightings” should be “a small number of sightings” (p. 15).

Corrected

“Tiger occupancy was based on surveys ... which was then spatially interpolated to non-survey areas” should be “Tiger occupancy was based on surveys ... which were then spatially interpolated to non-survey areas” (p. 19).

Well spotted. However, it is occupancy that was spatially interpolated, rather than the survey data. Therefore, we have corrected to:

‘Tiger occupancy, based on surveys implemented 2007 to 2009 34, was then spatially interpolated to non-surveyed areas.’

“physical & and population sizes” should be “physical and population sizes” (p. 21).

Corrected

“people an tigers” should be “people and tigers” (p. S6).

Corrected

References

Rossmo, D. K. (2000). Geographic profiling. Boca Raton, FL: CRC Press.

REVIEWERS' COMMENTS:

Reviewer #1 (Remarks to the Author):

Thank you for the opportunity to review the manuscript again. I have a few comments, attached. There are a number of grammatical issues with the manuscript, which could benefit from a very close copy edit.

[Editorial Note: See next page for Reviewer Comments.]

Summary of Comments on 1_reviewer_attachment_1_1529330000.pdf

Page: 3

- Number: 1 Author: Date: 6/18/2018 9:33:43 AM
Move this sentence to the end of the paragraph and perhaps revise to say. Consequently, research that explores links between E and S systems is mostly lacking although it remains a key priority for managers.

- Number: 2 Author: Date: 6/18/2018 9:34:24 AM
Be more specific. Pose a risk to human health and safety as well as livelihoods.

- Number: 3 Author: Date: 6/18/2018 9:34:44 AM
Perception of risk

- Number: 4 Author: Date: 6/18/2018 9:36:00 AM
The opening paragraph of the manuscript mentions coexistence, which I think is a good thing. Best to continue weaving that idea into the manuscript, like perhaps here.

Number: 1 Author: Date: 6/18/2018 9:36:51 AM

it is more than just a binary yes/no dimension of tolerance. Tolerance is nuanced, as you tell the reader below. Understanding and characterizing tolerance provides key insight about managing species.

 Number: 1 Author: Date: 6/18/2018 9:41:03 AM
Why not link back to tolerance and coexistence here?

Number: 1 Author: Date: 6/18/2018 9:48:19 AM

I find the tone of this sentence be prescriptive and patronizing. The authors are extremely accomplished individuals and scientists, but really, isn't this just a suggestion? Science is one part of the conservation equation, as I am sure all the authors all know. Perhaps revise so as to suggest that the evidence base clearly indicates space for integrated conceptual model—these models add nuanced color and focus to the picture that is previously lacking...or don't use a metaphor. But don't patronize....

Reviewer #2 (Remarks to the Author):

The revised study continues to be of scientific significance with important novelty that will advance the science around socio-ecological approaches to mitigating human-wildlife conflict, as well as practical conservation, management and law enforcement methods. The authors have addressed most of my concerns in this revision, but a few points remain as well as several new suggestions upon my reading of the revised text.

Additional follow-up needed for my comments from the previous review:

L424: Thank you for addressing my point regarding random points. The cited reference (47) doesn't present the discussion over the location of random points, and in fact this discussion hasn't appeared in the literature per se (it's more a discussion among colleagues outside of publications). Please delete "Although there is some debate regarding whether this background selection should be restricted⁴⁷" and start the sentence with the next phrase.

L357: I would appreciate if the authors could more fully address my question about whether measures of tolerance reflect people's behavior, and whether measures of tolerance are being interpreted equally by respondents with respect to how the posed questions would affect them (e.g. increase in tiger population in fact may or may not result in more conflict, but the survey assumes respondents think it will be a linear positive relationship). This question merits acknowledgement in the part of the Discussion about shortcomings of the study, because this is a fundamental assumption that we do not yet know to be true (whether all respondents are assuming the question to lead to the same response, and are answering accordingly) and to my knowledge there has not been research on this topic. My hope is that identifying this as a shortcoming in the study may prompt future research and exploration of this topic (perhaps even by the authors themselves).

New major suggestions:

Abstract: Despite the authors' minor changes to the abstract in an attempt to improve generalizability, the abstract as written does not reflect the full novelty of the study, which is in large part the new approach to socio-ecological data. Please bookend (add to the beginning and end) mention of the socio-ecological approach. Along with this, the sentences in L36-43 should be reorganized or rewritten to flow better to tell a story that builds into a final, generalizable lesson. Can you show how your results produce a general lesson, even while describing specific results? This would help solve Reviewer #1's original concern about generalizability of the study.

Title: The novelty of the study relates to the new socio-ecological framework but the title does not reflect this. I suggest revising the title to reflect the generalizable novelty of the study (the framework) rather than the specific case study (tigers; although keep 'tigers' in the title). This will also assist the reviewers' collective concerns about generalizability.

New minor suggestions:

L33: Please add a word or phrase to your first sentence after "poaching" to introduce conflict with people as a driver of tiger decline (e.g. "retaliatory killing" or "human-wildlife conflict"), since your first three sentences otherwise don't currently build on one another into a narrative. Habitat loss and poaching aren't necessarily related to human tolerance or 'managing wildlife', since these two drivers can result from other, unrelated causes. Mentioning retaliatory killing or human-wildlife conflict in the first sentence will enable you to lead into why coexisting with people would be relevant for predator conservation.

L34: Remove or refine the word "dangerous". Your first sentence mentions habitat loss and poaching but not threats to humans, so there's no set-up that the wildlife you're focusing on are "dangerous", which would be necessary to unpack this word, which is loaded with different meanings.

L55: "Many" is repetitive (also in L51); please reword.

L56: Change "public danger" to "danger to people".

L61: "encountered with large carnivores" – are you referring to encounters between people and carnivores or prey and carnivores? Here you appear to be discussing drivers of encounters between people and carnivores, but the analyzes you are referring to were to examine encounters between prey (both livestock and wild prey) and carnivores. Please find citations that refer to the landscape features associated with encounters between people (directly) and carnivores, or else rephrase to clarify that livestock-carnivore encounters lead to human-carnivore conflict and your specific point.

L91-92: As written, it's unclear whether this evidence suggests that the belief came (and led to) the action, or vice-versa. Did the study find that positive beliefs about wolf recovery were associated with (and in existence prior to) positive affective responses (which has a different meaning than responses leading to positive beliefs)? If not, I'm not confident this evidence supports your previous statement (that reaction comes from perception) and you may need to find different evidence.

L124-125: Change to "members of the Kerincinese and Minangkabau have lost kin" or "the Kerincinese and Minangkabau have lost members...". People (not groups) loose kin; groups loose members.

L159: Delete "fascinating" – not objective.

L190: Consider changing "retribution killings" to "retaliation killings", which is more commonly used with regards to human-wildlife conflict and more accurate (a response resulting from a cause, rather than a response aimed at revenge, which is insensitive to the livestock owners who are usually trying to prevent future attacks rather than pursue vengeance). Consider changing wherever mentioned in the article.

L241-242: Change 24.29 to 24 and 32.46 to 32 (round to nearest whole).

Table 1 and 2: Please define all covariate abbreviations somewhere, either in the footnotes or in the main text (and mention in the caption where the abbreviations are).

L269: Change to "In addition to age, ethnicity was also not selected as an important predictor in model selection (Table 2)..." What is meant by the latter part of the sentence? Please rephrase to be clear ("suggesting that connections with wildlife long thought to be associated specifically with Minangkabau and Kerincinese people did not permeate sufficiently to drive opinions towards tigers").

L286: Change "tigers" in "tiger numbers"

L288: Change "population" to "population size"

L311: Change semi-colon to comma.

L388-392: This revised text requires adjustment to increase clarity and the impact of its message. The

wording of this sentence is not entirely clear and misses an opportunity to better articulate expanding on the use and application of the framework. I suggest rephrasing the sentence as "Despite these shortcomings, our study suggests that tiger casualties and attacks on people may be averted or reduced by prioritizing responses or intervening proactively (e.g. via predator-proof livestock enclosures, compensation payments, and de-snaring patrols). We encourage conservation practitioners, natural resource managers and wildlife law enforcement to expand on and apply our framework to other socio-ecological systems globally to more fully explore its utility for facilitating human-wildlife coexistence."

L392-393: The mention about other species is not clear – delete or explain further.

Reviewer #3 (Remarks to the Author):

Many thanks for addressing my comments and queries so comprehensively. I am now recommending this paper is accepted for publication.

Reviewer #4 (Remarks to the Author):

The corrections are acceptable.
I recommend acceptance of the paper for publication.

NCOMMS-18-06811 (Tolerating tigers: Do local beliefs offset human-wildlife conflicts?)

Now titled: Addressing human-tiger conflict using socio-ecological information on tolerance and risk

RESPONSE TO REVIEWERS:

Reviewer #1 (Remarks to the Author):

Thank you for the opportunity to review the manuscript again. I have a few comments, attached. There are a number of grammatical issues with the manuscript, which could benefit from a very close copy edit.

... Comments from the pdf are copied below ...

Line 47: Move this sentence to the end of the paragraph and perhaps revise to say. Consequently, research that explores links between E and S systems is mostly lacking although it remains a key priority for managers. *We have combined the first two paragraphs to condense introductory material. We have retained the original position of the sentence, but have incorporated some of the reviewer suggestions:*

Line 48:

"Conservation science is often hindered by disciplinary boundaries¹. Consequently, despite benefits for management, research exploring links between ecological and social systems is limited¹⁻³"

Line 56: Be more specific. Pose a risk to human health and safety as well as livelihoods.

We have edited the sentence as follows:

Line 52:

"Situations involving mammalian carnivores exemplify this problem, as many are highly threatened, heavily persecuted, and pose a public threat¹¹."

Line 58: Perception of risk

We have retained the original wording in this revised paragraph in our effort to reduce the word count.

Line 68: The opening paragraph of the manuscript mentions coexistence, which I think is a good thing. Best to continue weaving that idea into the manuscript, like perhaps here.

In our shortened manuscript the terms 'coexistence' or 'coexist' now appear three times.

Line 70: It is more than just a binary yes/no dimension of tolerance. Tolerance is nuanced, as you tell the reader below. Understanding and characterizing tolerance provides key insight about managing species.

Absolutely. We agree and have edited the sentence accordingly:

Line 62:

"Understanding people's degree of tolerance for wildlife is key to managing dangerous and/or damage-causing species."

Line 132: Why not link back to tolerance and coexistence here?

Thank you for this suggestion. We have adopted this suggestion:

Line 104:

"These spiritual belief systems, coupled with on-going monitoring of tigers and their encounters with people, present a unique opportunity to investigate how these factors might foster tolerance and coexistence with dangerous wildlife."

Line 331: I find the tone of this sentence prescriptive and patronizing. The authors are extremely accomplished individuals and scientists, but really, isn't this just a suggestion? Science is one part of the conservation equation, as I am sure all authors all know. Perhaps revise so as to suggest that the evidence base clearly indicates space for integrated conceptual models – these models add nuanced color and focus to the picture that is previously lacking...or don't use a metaphor. But don't patronize....

This is a good point, and we apologise for the unintended tone. We have edited the sentence in question to:

Line 245:

“Evidence suggests that applying socio-ecological models to conservation conflicts can be informative and beneficial; such models are rare, but have been increasingly applied in recent years⁵.”

Reviewer #2 (Remarks to the Author):

The revised study continues to be of scientific significance with important novelty that will advance the science around socio-ecological approaches to mitigating human-wildlife conflict, as well as practical conservation, management and law enforcement methods. The authors have addressed most of my concerns in this revision, but a few points remain as well as several new suggestions upon my reading of the revised text.

Additional follow-up needed for my comments from the previous review:

L424: Thank you for addressing my point regarding random points. The cited reference (47) doesn't present the discussion over the location of random points, and in fact this discussion hasn't appeared in the literature per se (it's more a discussion among colleagues outside of publications). Please delete “Although there is some debate regarding whether this background selection should be restricted⁴⁷” and start the sentence with the next phrase.

We have deleted the sentence identified and the associated reference [47].

L357: I would appreciate if the authors could more fully address my question about whether measures of tolerance reflect people's behavior, and whether measures of tolerance are being interpreted equally by respondents with respect to how the posed questions would affect them (e.g. increase in tiger population in fact may or may not result in more conflict, but the survey assumes respondents think it will be a linear positive relationship). This question merits acknowledgement in the part of the Discussion about shortcomings of the study, because this is a fundamental assumption that we do not yet know to be true (whether all respondents are assuming the question to lead to the same response, and are answering accordingly) and to my knowledge there has not been research on this topic. My hope is that identifying this as a shortcoming in the study may prompt future research and exploration of this topic (perhaps even by the authors themselves).

We feel that the previous addition made to the manuscript adequately and concisely addresses the concerns raised by the reviewer so we have refrained from expanding this part of the discussion further.

Line 271:

“Measures of tolerance may better reflect people's actual ability to coexist with a species if linked explicitly to an effect, for example, the probability of increased human-tiger interaction given a specified population increase. However, data to specify such statements are lacking in most human-wildlife systems.”

New major suggestions:

Abstract: Despite the authors' minor changes to the abstract in an attempt to improve generalizability, the abstract as written does not reflect the full novelty of the study, which is in large part the new approach to socio-ecological data. Please bookend (add to the beginning and end) mention of the socio-ecological approach. Along with this, the sentences in L36-43 should be reorganized or rewritten to flow better to tell a story that builds into a final, generalizable lesson. Can you show how your results produce a general lesson, even while describing specific results? This would help solve Reviewer #1's original concern about generalizability of the study.

We have revised the abstract to address these concerns. Now that we have included 'socio-ecological' in the title we have avoided over-use of the term in the abstract, and so have not bookended the term as

suggested by the reviewer. However, we have made it clear that the novelty is the socio-ecological information and interdisciplinary nature of the study:

*“Tigers are critically endangered due to deforestation and persecution. Yet in places, Sumatran tigers (*Panthera tigris sumatrae*) continue to coexist with people, offering insights for managing wildlife elsewhere. Here, we couple spatial models of encounter risk with information on tolerance from 2,386 Sumatrans to reveal drivers of human-tiger conflict. Risk of encountering tigers was greater around populated villages that neighbored forest or rivers connecting tiger habitat; geographic profiles refined these predictions to three core areas. People’s tolerance for tigers was related to underlying attitudes, emotions, norms and spiritual beliefs. Combining this information into socio-ecological models yielded predictions of tolerance that were 32 times better than models based on social predictors alone. Pre-emptive intervention based on these socio-ecological predictions could have averted up to 51% of attacks on livestock and people, saving 15 tigers from the wild. Our work provides further evidence of the benefits of interdisciplinary research on conservation conflicts.”*

Title: The novelty of the study relates to the new socio-ecological framework but the title does not reflect this. I suggest revising the title to reflect the generalizable novelty of the study (the framework) rather than the specific case study (tigers; although keep ‘tigers’ in the title). This will also assist the reviewers’ collective concerns about generalizability.

*Taking this comment on board and complying to journal style we have revised the title to:
“Addressing human-tiger conflict using socio-ecological information on tolerance and risk”*

New minor suggestions:

L33: Please add a word or phrase to your first sentence after “poaching” to introduce conflict with people as a driver of tiger decline (e.g. “retaliatory killing” or “human-wildlife conflict”), since your first three sentences otherwise don’t currently build on one another into a narrative. Habitat loss and poaching aren’t necessarily related to human tolerance or ‘managing wildlife’, since these two drivers can result from other, unrelated causes. Mentioning retaliatory killing or human-wildlife conflict in the first sentence will enable you to lead into why coexisting with people would be relevant for predator conservation.

In addition to other edits made throughout the Abstract (presented in full above), we have revised the word ‘poaching’ to ‘persecution’ in order to capture different types of tiger-killing by humans (e.g. retaliatory killing or poaching). The tight word limit for the abstract constraints us from explicitly listing poaching, human wildlife conflict and retaliatory killings, but we would welcome feedback on how this could be achieved while keeping the remaining messages.

L34: Remove or refine the word “dangerous”. Your first sentence mentions habitat loss and poaching but not threats to humans, so there’s no set-up that the wildlife you’re focusing on are “dangerous”, which would be necessary to unpack this word, which is loaded with different meanings.

We have chosen to remove the word ‘dangerous’.

L55: “Many” is repetitive (also in L51); please reword.

We have revised this entire paragraph and addressed this repetition

L56: Change “public danger” to “danger to people”.

In our shortened Introduction this now reads ‘public threat’.

L61: “encountered with large carnivores” – are you referring to encounters between people and carnivores or prey and carnivores? Here you appear to be discussing drivers of encounters between people and carnivores, but the analyzes you are referring to were to examine encounters between prey (both livestock and wild prey) and carnivores. Please find citations that refer to the landscape features associated with encounters between people (directly) and carnivores, or else rephrase to clarify that livestock-carnivore encounters lead to human-carnivore conflict and your specific point.

We have removed this sentence in the shortened Introduction.

L91-92: As written, it's unclear whether this evidence suggests that the belief came (and led to) the action, or vice-versa. Did the study find that positive beliefs about wolf recovery were associated with (and in existence prior to) positive affective responses (which has a different meaning than responses leading to positive beliefs)? If not, I'm not confident this evidence supports your previous statement (that reaction comes from perception) and you may need to find different evidence.

We have removed reference to this study in the shortened Introduction.

L124-125: Change to "members of the Kerincinese and Minangkabau have lost kin" or "the Kerincinese and Minangkabau have lost members...". People (not groups) loose kin; groups loose members.

Changed to:

Line 99:

"For example, the Kerincinese and Minangkabau people..."

L159: Delete "fascinating" – not objective.

We have deleted 'fascinating' as suggested.

L190: Consider changing "retribution killings" to "retaliation killings", which is more commonly used with regards to human-wildlife conflict and more accurate (a response resulting from a cause, rather than a response aimed at revenge, which is insensitive to the livestock owners who are usually trying to prevent future attacks rather than pursue vengeance). Consider changing wherever mentioned in the article.

We used the phrase retribution killings twice in the manuscript. Both instances have now been edited to retaliatory killings.

L241-242: Change 24.29 to 24 and 32.46 to 32 (round to nearest whole).

Done

Table 1 and 2: Please define all covariate abbreviations somewhere, either in the footnotes or in the main text (and mention in the caption where the abbreviations are).

Covariates abbreviates are all provided in the Methods section, and also in the SI. Now that we have reformatted the Table legends to closely match the journal style we have explicitly referred the reader to these sections.

L269: Change to "In addition to age, ethnicity was also not selected as an important predictor in model selection (Table 2)..." What is meant by the latter part of the sentence? Please rephrase to be clear ("suggesting that connections with wildlife long thought to be associated specifically with Minangkabau and Kerincinese people did not permeate sufficiently to drive opinions towards tigers").

Sorry for the confusion. We have edited this section to:

Line 206:

"In addition to age, ethnicity was not selected in our models as an important predictor of people's tolerance towards tigers (Table 2). Rather, underlying psychological factors, including attitudes, human emotion, and beliefs associated with overall spiritual wellbeing, were the strongest significant predictors of people's connections with tigers overall, as evidenced by large model-averaged β coefficients and variable importance values (Table 2)."

L286: Change "tigers" in "tiger numbers"

Done

L288: Change "population" to "population size"

Done

L311: Change semi-colon to comma.

Done

L388-392: This revised text requires adjustment to increase clarity and the impact of its message. The wording of this sentence is not entirely clear and misses an opportunity to better articulate expanding on the use and application of the framework. I suggest rephrasing the sentence as “Despite these shortcomings, our study suggests that tiger casualties and attacks on people may be averted or reduced by prioritizing responses or intervening proactively (e.g. via predator-proof livestock enclosures, compensation payments, and de-snaring patrols). We encourage conservation practitioners, natural resource managers and wildlife law enforcement to expand on and apply our framework to other socio-ecological systems globally to more fully explore its utility for facilitating human-wildlife coexistence.”

We have incorporated this suggestion, as follows:

Line 302:

“Despite these shortcomings, our study suggests that attacks by and towards tigers could be averted or reduced by prioritizing responses or intervening proactively (e.g. via predator-proof livestock enclosures, compensation payments, and de-snaring patrols). We encourage conservation practitioners to expand upon and apply our framework to other socio-ecological systems globally to more fully explore its utility for facilitating human-wildlife coexistence.”

L392-393: The mention about other species is not clear – delete or explain further.

We have deleted this sentence as suggested.

Reviewer #3 (Remarks to the Author):

Many thanks for addressing my comments and queries so comprehensively. I am now recommending this paper is accepted for publication.

Reviewer #4 (Remarks to the Author):

The corrections are acceptable.

I recommend acceptance of the paper for publication.

References

Redpath SM, Gutiérrez RJ, Wood KA, Young JC. 2015. Conflicts in conservation: Navigating towards solutions. Cambridge University Press, Cambridge, UK.